# ON THE INFLATION OF KNN-SHAPLEY VALUE

## ABSTRACT

Shapley value-based data valuation methods, originating from cooperative game theory, quantify the usefulness of each individual sample by considering its contribution to all possible training subsets. Despite their extensive applications, we observe these methods encounter value inflation—while samples with negative Shapley values are detrimental, some with positive values can also be harmful. This challenge prompts two fundamental questions: the suitability of zero as a threshold for distinguishing detrimental from beneficial samples and the determination of an appropriate threshold. To address these questions, we focus on KNN-Shapley and propose Calibrated KNN-Shapley (CKNN-Shapley), a semi-value method that calibrates zero as the threshold to distinguish detrimental samples from beneficial ones by mitigating the negative effects of small-sized training subsets. Through extensive experiments, we demonstrate the effectiveness of CKNN-Shapley in alleviating data valuation inflation, detecting detrimental samples, and assessing data quality. We also extend our approach beyond conventional classification settings, applying it to diverse and practical scenarios such as learning with mislabeled data, online learning with stream data, and active learning for label annotation.

## 1 INTRODUCTION

The significance of data as inputs in machine learning algorithms cannot be overstated for algorithmic performance (Liang et al., 2022; Zha et al., 2023). Beyond the development of sophisticated algorithms, there is a growing emphasis on the curation of high-quality training sets. Data-centric learning has emerged to assess the valuation of data within the context of a learning task. This encompasses a spectrum of tasks, ranging from outlier detection (Boukerche et al., 2020) to noisy label correction (Zheng et al., 2021), from best subset selection (Hazimeh and Mazumder, 2020) to sample reweighting (Li and Liu, 2022), and antidote data generation (Chhabra et al., 2022; Li et al., 2023) to active labeling (Tharwat and Schenck, 2023; Liu et al., 2021).

Understanding the value of an individual data sample is fundamental in data-centric learning. Leave-one-out influence (Cook and Weisberg, 1982), a straightforward method, offers an initial assessment of the relative influence of the specific sample compared to the rest of the training set. Shapley value-based methods (Shapley et al., 1953; Roth, 1988) quantify the usefulness of each individual sample towards the utility on a validation set by considering its contribution to all possible training subsets. Unlike the leave-one-out influence, Shapley value represents the weighted average utility change resulting from adding the point to different training subsets, showcasing greater robustness in diverse contexts (Bordt and von Luxburg, 2023; Tsai et al., 2023; Sundararajan et al., 2020). Despite the absence of assumptions on the learning model, Shapley value-based methods require expensive and extensive model retraining, which are prohibitive for large-scale data analysis and deep models (Hammoudeh and Lowd, 2022).

With the advent of KNN-Shapley (Jia et al., 2019b; 2021), a pragmatic tool enabling the computation of Shapley values without the need for costly model retraining, Shapley-based approaches have become feasible and widely applied. KNN-Shapley leverages the K-Nearest Neighbors (KNN) classifier as a surrogate for the original learning model, recursively calculating the Shapley value for each training sample. Despite their promise, KNN-Shapley and its variants grapple with the issue of value inflation. We note that While samples with negative Shapley values are recognized as detrimental, the dilemma arises as certain samples with positive values may also have harmful effects. This challenge gives rise to two pivotal questions: the appropriateness of zero as a threshold for distinguishing detrimental from beneficial samples and the identification of a suitable threshold.

**Our Contributions.** In this paper, we focus on addressing the value inflation issue observed in KNN-Shapley. To tackle this, we propose Calibrated KNN-Shapley (CKNN-Shapley), a semi-value method that calibrates zero as the threshold to distinguish detrimental samples from beneficial ones by mitigating negative effects of small training subsets. Our contributions are summarized as follows:

- We unveil the value inflation issue of KNN-Shapley, which not only misidentifies a portion of detrimental samples as beneficial, but also distorts the interpretation of data valuation. Beyond the misidentified samples, value inflation further impacts the assessment of beneficial samples.

- We propose CKNN-Shapley, a simple but not incremental method to calibrate zero as the threshold to distinguish detrimental samples from beneficial ones. Our hypothesis attributes value inflation to improper subset selection in KNN-Shapley, and we address this issue through a straightforward yet effective strategy by imposing a constraint on the size of training subsets.

- We perform comprehensive experiments on various benchmark datasets, comparing CKNN-Shapley with KNN-Shapley-based methods. The results showcase the effectiveness of CKNN-Shapley in mitigating value inflation and improving classification performance. We also extend our approach beyond conventional classification settings, applying it to learning with mislabeled data, online learning with stream data, and active learning for label annotation.

**Related Work**. We introduce Shapley-based methods with a major focus on data valuation and KNN-Shapley. *(i) Shapley-based Data Valuation*. The Shapley value (Shapley et al., 1953; Roth, 1988) measures the weighted average utility change when adding a point to all possible training subsets, making it a primary tool for assessing the valuation of individual samples (Jiang et al., 2023). Shapley-value based methods have found extensive application in various domains, including variable selection (Cohen et al., 2005; Zaeri-Amirani et al., 2018), feature importance (Lundberg and Lee, 2017; Covert and Lee, 2020; Jethani et al., 2021), model valuation (Rozemberczki and Sarkar, 2021), health care (Pandl et al., 2021; Tang et al., 2021), federated learning (Han et al., 2021), collaborative learning (Sim et al., 2020), data debugging (Deutch et al., 2021), and distribution analysis (Schoch et al., 2022; Ghorbani et al., 2020). Building upon this concept, Beta Shapley (Kwon and Zou, 2022) and Banzhaf value (Wang and Jia, 2023a) have been developed by relaxing the efficiency axiom of the Shapley value. While Shapley-based data valuation approaches are model-agnostic, exponential model retraining renders these methods computationally challenging even for small datasets (Ghorbani and Zou, 2019; Jia et al., 2019b). Efforts to accelerate computation include efficient sampling (Zhang et al.), utility learning (Wang et al., 2021), and the assumption of independent utility (Luo et al., 2022). In addition to data valuation. *(ii) KNN-Shapley Data Valuation*. KNN-Shapley (Jia et al., 2019a; Wang and Jia, 2023b) emerges as one of the most promising solutions to mitigate the computational challenges associated with Shapley values. It leverages assumptions about the learning model by employing a KNN classifier as a surrogate, recursively computing the Shapley value for each training sample without the need for retraining. This high efficiency has spurred the development of numerous KNN-Shapley variants, such as weighted KNN (Wang et al., 2024), soft KNN (Wang and Jia, 2023b), and threshold KNN(Wang et al., 2023), tailored to enhance generalization, sample reuse, and privacy risks, respectively. In this paper, our focus is on the KNN-Shapley category, with the objective of addressing issues related to value inflation.

## 2 PRELIMINARIES AND MOTIVATION

**Preliminaries**. For a training set $\mathcal{D}$ with $N$ samples and a learning algorithm $\mathcal{A}$, let $U_{\mathcal{A},\mathcal{D}_v}(\mathcal{D})$ represent the model utility with all training data on the validation set $\mathcal{D}_v$. For simplicity, we use $U(\mathcal{D})$ in the following manuscript. The Shapley value (Shapley et al., 1953) of a training sample $z_i \in D$, $1 \leq i \leq N$ is defined as follows:

$$\nu^s(z_i) = \frac{1}{N} \sum_{\mathcal{S} \subseteq \mathcal{D} \setminus z_i} \frac{1}{\binom{N-1}{|\mathcal{S}|}} [U(\mathcal{S} \cup z_i) - U(\mathcal{S})], \tag{1}$$

where $\mathcal{S}$ is a training subset. The Shapley value gauges the average contribution of $z_i$ on all possible subsets of $\mathcal{D}$ without $z_i$ from the cooperative game perspective. Note that obtaining the exact Shapley value necessitates $2^N$ model training. Despite no assumption of the learning algorithm $\mathcal{A}$, for large models, this process consumes significant computational resources and time. The impracticality of calculating the exact Shapley value, even using Monte Carlo approximation, becomes evident in large-scale data analyses involving substantial models due to its high computational complexity.

KNN-Shapley (Jia et al., 2019b) occurs as a pragmatic tool for computing Shapley values efficiently. It employs a KNN classifier as a surrogate for the learning algorithm $\mathcal{A}$. For a single validation sample $\mathcal{D}_v = \{z_v\}$, where $z_v = (x_v, y_v)$ contains $x_v$ in the feature space and $y_v$ in the label space, a KNN classifier sorts the whole training data and identifies $K$ nearest neighbors in the feature space $(z_{\alpha_1}, z_{\alpha_2}, \cdots, z_{\alpha_K})$ to $z_v$, where $\alpha_k$ represents the index of the training samples with the $k$-th neighbor to the validation sample $z_v$. If the predictive confidence is used as the model utility, *i.e.*, $U(\mathcal{S}) = \frac{1}{K} \sum_{k=1}^{\min\{K, |\mathcal{S}|\}} \mathbb{1}[y_{\alpha_k} = y_v]$, KNN-Shapley values can be calculated recursively as follows:

$$\nu^k(z_{\alpha_N}) = \frac{\mathbb{1}[y_{\alpha_N} = y_v]}{N}, \ \ \nu^k(z_{\alpha_i}) = \nu^k(z_{\alpha_{i+1}}) + \frac{\mathbb{1}[y_{\alpha_i} = y_v] - \mathbb{1}[y_{\alpha_{i+1}} = y_v]}{\max\{K, i\}}. \quad (2)$$

The above Eq. (2) can be extended to multiple validation samples by summing up the KNN-Shapley value for each validation sample. This results in a time complexity of $\mathcal{O}(N \log N)$ for KNN-Shapley, significantly faster than the vanilla Shapley. It is worth noting that KNN is a lazy classifier without any training process; therefore, within the retrain-based Shapley framework, KNN-Shapley can directly conduct the inference process, significantly reducing the time complexity compared to other training-required classifiers. Moreover, for certain deep models, KNN-Shapley is compatible with the embedding of training/validation samples for data valuation (Jia et al., 2019b).

**Inflation of KNN-Shapley Value**. Despite the wide range of applications and efficiency of KNN-Shapley, we have observed a phenomenon of value inflation during its practical usage. This occurs when, contrary to the expectation that samples with negative Shapley values are detrimental, some samples with positive values can also be harmful. Figure 1 illustrates this phenomenon on the *SST-2* dataset (Socher et al., 2013). The blue bar plot shows the histogram of KNN-Shapley values for training samples. According to the ascending order of their values, we segment the whole training set into 20 equally sized bins (which are different from the histogram bins). To identify whether samples in a specific bin are detrimental or beneficial, we train the KNN classifier with that bin removed from the training set (denoted by the red line) and compare its performance with the complete training set (denoted by the green line). Due to the equal-size samples in each bin, the markers on the red line do not have uniform intervals and do not align with unevenly sized histograms. While samples with negative KNN-Shapley values are generally detrimental, the highlighted green shallow region reveals a noteworthy observation—samples in this region are harmful to the learning task despite having positive KNN-Shapley values, indicating the issue of KNN-Shapley value inflation. Note that the number of samples with negative KNN-Shapley values is 1,452, while 4,548 samples are in the misidenti-

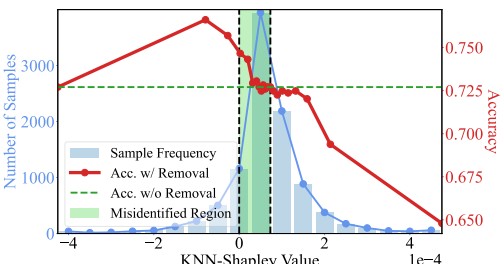

Figure 1: Illustration of KNN-Shapley value inflation. The **blue bar plot** with x-axis and blue left y-axis displays the histogram of KNN-Shapley values for training samples in the *SST-2* dataset (Socher et al., 2013). For the purpose of visualization, we merge the samples with extremely small or large values into the leftmost or rightmost bars. With another segmentation of 20 equally-sized bins based on the ascending order of their values, the **red line** illustrates KNN performance (in red right y-axis) with **a specific bin** (with average Shapley value in x-axis) removed from the training set, while the dashed green line represents KNN performance on the entire training set. By comparing the red and green lines, the detrimental bin can be identified, as the performance improves upon its removal. While samples with negative KNN-Shapley values are detrimental, a notable observation is the green shallow highlighted region, where samples are harmful to the learning task despite having positive KNN-Shapley values, indicating the issue of KNN-Shapley value inflation.

fied region. In our later experiments (Table 2), removing all the samples with negative KNN-Shapley values enhances performance from 0.7270 to 0.8160. Conversely, further improvement is achieved by removing samples in the misidentified region, boosting performance from 0.8160 to 0.8980 and underscoring the significance of addressing inflated samples.

Value inflation prompts two fundamental questions: the suitability of zero as a threshold for distinguishing detrimental from beneficial samples and the determination of an appropriate threshold. To address these questions, we propose Calibrated KNN-Shapley in the next section, which calibrates zero as the threshold to distinguish detrimental samples from beneficial ones.

## 3 CALIBRATED KNN-SHAPLEY

To address the challenge of valuation inflation in KNN-Shapley, we first delve into its underlying mechanism and scrutinize potential factors contributing to negative effects on data valuation. Based on the analyzed reasons, we propose our Calibrated KNN-Shapley by mitigating the negative efforts from improper training subsets.

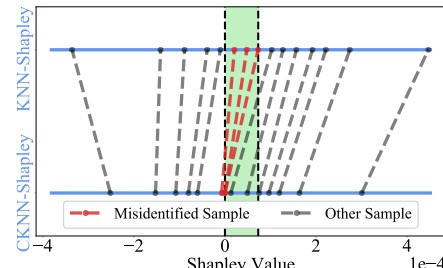

While the recursive formulation in Eq. (2) brings about efficient computation, it also introduces inevitable accumulated errors. Specifically, the valuation of a training sample distant from the validation sample can influence other training samples closer to the validation sample, emphasizing the importance of the first term $\nu^k(z_{\alpha_N})$. Upon closer examination of $z_{\alpha_N}$, the farthest sample from the validation sample, we observe its minimal impact on the validation sample. Considering the definition of the Shapley value, which measures the average contribution of a training sample across all possible subsets, we discuss two cases. Case I: when $|\mathcal{S}|>K$, $z_{\alpha_N}$ does not contribute to the utility of the KNN classifier since it is not among the neighbors of $z_v$. Case II: when $|\mathcal{S}|\leq K$, $U(S \cup z_{\alpha_N}) - U(S)$ is non-negative, *i.e.*, $U(S \cup z_{\alpha_N}) - U(S){=}1/N$

Figure 2: Comparison of KNN-Shapley and CKNN-Shapley value on the *SST-2* dataset, where each dashed line represents a training sample associated with its KNN-Shapley and CKNN-Shapley values, respectively, and the green region is the misidentified detrimental samples from Figure 1. The red dashed lines denote the samples that are incorrectly identified by KNN-Shapley but correctly identified by CKNN-Shapley.

if $z_{\alpha_N}$ and $z_v$ share the same label; otherwise, $U(S \cup z_{\alpha_N}) - U(S){=}0$. Notably, it is impractical and meaningless to have a training subset with only a few training samples. Furthermore, the number of occurrences in Case I significantly surpasses that in Case II, indicating that Case II is not representative of learning models and data valuation.

Building on the above analyses, we hypothesize that value inflation stems from improper subset selection in KNN-Shapley. Certain subsets with only a few samples exhibit significant divergence from the original set, leading to an exaggeration of the contribution of a specific sample on these subsets. This, in turn, gives rise to the phenomenon of value inflation. To address the issue of value inflation, we introduce Calibrated KNN-Shapley (CKNN-Shapley) through the selection of suitable training subsets. The training subset in CKNN-Shapley should serve as an effective proxy for the original and complete training set. In this paper, we present a straightforward yet effective strategy by imposing a constraint on the size of training subsets, specifically $|\mathcal{S}|\geq T$, where $T$ represents the size of the smallest training subsets used to assess the contribution of each training sample. This strategy is a semi-value, which is computed as follows:[1]

$$\nu^c(z_{\alpha_N}) = \nu^c(z_{\alpha_{N-1}}) = \cdots = \nu^c(z_{\alpha_{N-T+1}}) = 0,$$

$$\nu^c(z_{\alpha_{N-T}}) = \frac{\mathbb{1}[y_{\alpha_{N-T}} = y_v]}{N - T}, \ \nu^c(z_{\alpha_i}) = \nu^c(z_{\alpha_{i+1}}) + \frac{\mathbb{1}[y_{\alpha_i} = y_v] - \mathbb{1}[y_{\alpha_{i+1}} = y_v]}{\max\{K, i\}}. \tag{3}$$

Compared to KNN-Shapley in Eq. (2), CKNN-Shapley follows a similar recursive fashion but more efficiently. This efficiency stems from directly assigning zero to $T$ samples that are far away from the validation sample. Additionally, in CKNN-Shapley, the selected training subsets consist of at least $T$ samples, preventing the inclusion of improper subsets that could contribute to the valuation.

Figure 2 illustrates the comparison between KNN-Shapley and CKNN-Shapley values on the *SST-2* dataset. Each dashed line represents a training sample associated with its KNN-Shapley and CKNN-Shapley values. Detrimental samples misidentified by KNN-Shapley, highlighted by red dashed lines, are assigned negative or zero values in CKNN-Shapley, suggesting that zero in CKNN-Shapley serves as a suitable threshold for distinguishing detrimental from beneficial samples. Additionally, the majority of lines from KNN-Shapley to CKNN-Shapley move from the top right to the bottom left. This indicates that KNN-Shapley not only has a negative effect on the misidentified samples but also tends to inflate the valuation of most samples. This inflation might be attributed to its recursive formulation, leading to the accumulation of errors within KNN-Shapley.

---

[1]Another interpretation of Eq. (3) is provided in Appendix A.

Table 1: Threshold for distinguishing detrimental from beneficial samples and the misidentification ratio of detrimental samples in Eq. (4) for KNN-Shapley-based methods. For the first metric, closer to zero is preferable; Regarding the second one, smaller values suggest better performance. The last column takes the absolute values for average. The best results are highlighted in bold. TKNN-Shapley does not return meaningful results on *CIFAR10* with the default parameters, denoted by "N/A."

| Datasets | MNIST | FMNIST | CIFAR10 | Pol | Wind | CPU | AGnews | SST-2 | News20 | Avg. |
|---|---|---|---|---|---|---|---|---|---|---|
| Threshold for distinguishing detrimental from beneficial samples by $\times 1e-4$ | | | | | | | | | | |
| KNN-Shapley 2019a | 0.2744 | 0.2631 | 0.1632 | 2.0361 | 7.3705 | 6.4458 | 1.6264 | 1.3659 | 1.7215 | 2.3630 |
| KNN-Shapley-JW 2023b | 0.2544 | 0.2431 | 0.1432 | 0.5119 | 4.6543 | 3.7429 | 1.3763 | 0.8203 | 1.6780 | 1.4916 |
| TKNN-Shapley 2023 | 3.8641 | 1.0330 | N/A | -1.4990 | 1.0875 | 5.6960 | 1.1951 | 0.7143 | 5.2628 | 2.5440 |
| KNN-Beta Shapley 2022 | **0.0149** | 0.0317 | **0.0739** | 10.1847 | 8.0875 | 9.5856 | **0.3239** | 1.4949 | **0.0480** | 3.2504 |
| CKNN-Shapley (Ours) | 0.0152 | **0.0109** | 0.1163 | **-0.3059** | **-0.0245** | **-2.9412** | 1.1955 | **0.1604** | 0.6715 | **0.6046** |
| Misidentification ratio of detrimental samples | | | | | | | | | | |
| KNN-Shapley 2019a | 0.8065 | 0.5145 | 0.4908 | 0.3333 | 0.3864 | 0.5714 | 0.7580 | 0.5600 | 0.6179 | 0.5599 |
| KNN-Shapley-JW 2023b | 0.7665 | 0.4367 | 0.3792 | 0.0000 | 0.2727 | 0.3846 | 0.6563 | 0.3421 | 0.5541 | 0.4214 |
| TKNN-Shapley 2023 | 0.3103 | 1.0000 | N/A | **0.0000** | 0.1818 | 0.4118 | 0.2273 | 0.1862 | 0.2000 | 0.3147 |
| KNN-Beta Shapley 2022 | 1.0000 | 1.0000 | 1.0000 | 1.0000 | 1.0000 | 1.0000 | 1.0000 | 1.0000 | 1.0000 | 1.0000 |
| CKNN-Shapley (Ours) | **0.2000** | **0.1686** | **0.0908** | **0.0000** | **0.0000** | **0.0000** | **0.1538** | **0.1143** | **0.1821** | **0.1011** |

Table 2: Classification performance of KNN-Shapley-based methods. # denotes KNN's performance trained on the training set excluding samples with negative valuations, while + represents weighted KNN's performance, where the weights are derived from data valuations.

| Method\Datasets | MNIST | FMNIST | CIFAR10 | Pol | Wind | CPU | AGnews | SST-2 | News20 | Avg. |
|---|---|---|---|---|---|---|---|---|---|---|
| Vanilla KNN | 0.9630 | 0.8444 | 0.5956 | 0.9400 | 0.8700 | 0.9200 | 0.9060 | 0.7270 | 0.6920 | 0.8287 |
| KNN-Shap 2019a# | 0.9682 | 0.8586 | 0.6456 | **0.9700** | 0.8750 | 0.9650 | 0.9250 | 0.8160 | 0.7580 | 0.8646 |
| KNN-Shap-JW 2023b# | 0.9698 | 0.8574 | 0.6514 | **0.9700** | 0.8650 | 0.9600 | 0.9250 | 0.8498 | 0.7610 | 0.8677 |
| TKNN-Shap 2023# | 0.6644 | 0.8356 | N/A | 0.8150 | 0.8300 | 0.9100 | 0.8990 | 0.7982 | 0.6730 | 0.8032 |
| KNN-Beta Shap 2022# | 0.9630 | 0.8444 | 0.5956 | 0.9400 | 0.8700 | 0.9200 | 0.9060 | 0.7270 | 0.6920 | 0.8287 |
| CKNN-Shap (Ours)# | **0.9828** | **0.8884** | **0.7164** | **0.9700** | **0.9000** | **0.9700** | **0.9420** | **0.8980** | **0.7960** | **0.8960** |
| KNN-Shap 2019a+ | 0.9742 | 0.8680 | 0.6598 | 0.9650 | 0.8700 | 0.9300 | 0.9333 | 0.8480 | 0.7790 | 0.8697 |
| KNN-Shap-JW 2023b+ | 0.9754 | 0.8658 | 0.6610 | 0.9650 | 0.8700 | 0.9300 | 0.9320 | 0.8555 | 0.7790 | 0.8704 |
| TKNN-Shap 2023+ | 0.8626 | 0.7900 | N/A | 0.8600 | 0.8200 | 0.8950 | 0.9110 | 0.8326 | 0.7160 | 0.8359 |
| KNN-Beta Shap 2022+ | 0.9654 | 0.8490 | 0.6056 | **0.9750** | 0.8850 | 0.9450 | 0.9120 | 0.7305 | 0.7140 | 0.8424 |
| CKNN-Shap (Ours)+ | **0.9890** | **0.9106** | **0.7404** | 0.9700 | **0.9050** | **0.9650** | **0.9450** | **0.8920** | **0.8130** | **0.9033** |

As a variant of KNN-Shapley, our CKNN-Shapley inherently integrates several key axioms from KNN-Shapley, focusing on three fundamental principles: group rationality, fairness, and additivity, as emphasized in (Jia et al., 2019b). CKNN-Shapley prioritizes fairness, ensuring symmetry (where two identical samples receive identical Shapley values) and zero elements (no contribution, no payment). Additionally, it upholds additivity, where values across multiple utilities sum up to the value under a utility that is the aggregate of all these utilities. Enhancing group rationality, CKNN-Shapley refines this axiom by disregarding subsets with fewer than $T$ samples, thereby distributing the utility difference between the entire dataset and the subset. Since the utility of a subset, according to the KNN classifier, expects only the nearest samples to contribute, the utility of such a subset is anticipated to be zero, a factor overlooked in KNN-Shapley. By addressing this, CKNN-Shapley mitigates the adverse impact of overemphasizing the cooperative game with samples distant from the target sample, rendering the axiom of group rationality more compatible with the KNN classifier.

## 4 EXPERIMENTAL RESULTS

**Experimental Setup**. We evaluate our method on nine datasets: *MNIST*, *FMNIST* (Xiao et al., 2017), *CIFAR10*, *Pol*, *Wind*, *CPU* (Wang et al., 2023), *AGnews*, *SST-2* (Socher et al., 2013), and *News20* (Lang, 1995). For *CIFAR10* and the text datasets, we use ResNet50 (He et al., 2016) and Sentence Bert (Reimers and Gurevych, 2019) embeddings. Baseline methods include KNN-Shapley (Jia et al., 2019a), KNN-Shapley-JW (Wang and Jia, 2023b), TKNN-Shapley (Wang et al., 2023), KNN-Beta Shapley (Kwon and Zou, 2022), and our CKNN-Shapley.[2] See Appendix B for details on datasets, baseline methods and the experimental environment.

In addition to the conventional predictive accuracy for classification, we introduce two metrics to assess valuation inflation: the threshold for distinguishing detrimental samples from beneficial ones and the misidentification ratio of detrimental samples. This approach is inspired by the illustration in

---

[2]Our code is available at: https://anonymous.4open.science/r/Inflation_KNN-SV-D9C1.

Table 3: Classification performance of KNN-Shapley-based methods with fixed numbers of removed samples. #, +, and ∗ denote KNN's performance trained on the training set excluding samples with the smallest 10%, 20%, and 30% data valuations, respectively.

| Method\Datasets | MNIST | FMNIST | CIFAR10 | Pol | Wind | CPU | AGnews | SST-2 | News20 | Avg. |
|---|---|---|---|---|---|---|---|---|---|---|
| Vanilla KNN | 0.9630 | 0.8444 | 0.5956 | 0.9400 | 0.8700 | 0.9200 | 0.9060 | 0.7270 | 0.6920 | 0.8287 |
| KNN-Shap 2019a# | 0.9702 | 0.8590 | 0.6270 | **0.9750** | 0.8850 | 0.9550 | 0.9250 | 0.8016 | 0.7290 | 0.8585 |
| KNN-Shap-JW 2023b# | 0.9702 | 0.8590 | 0.6270 | 0.9700 | 0.8850 | 0.9550 | 0.9250 | 0.8016 | 0.7280 | 0.8579 |
| TKNN-Shap 2023# | 0.9516 | 0.8368 | 0.5670 | 0.9450 | 0.8450 | 0.9200 | 0.9190 | 0.7878 | 0.7160 | 0.8320 |
| KNN-Beta Shap 2022# | 0.9650 | 0.8486 | 0.6072 | 0.9600 | **0.8950** | 0.9350 | 0.9130 | 0.7408 | 0.7210 | 0.8428 |
| CKNN-Shap (Ours)# | **0.9812** | **0.8822** | **0.6566** | 0.9650 | **0.8950** | **0.9700** | **0.9370** | **0.8108** | **0.7450** | **0.8714** |
| KNN-Shap 2019a+ | 0.9714 | 0.8548 | 0.6396 | 0.9650 | 0.8650 | 0.9550 | 0.9250 | 0.8314 | 0.7460 | 0.8615 |
| KNN-Shap-JW 2023b+ | 0.9714 | 0.8548 | 0.6396 | 0.9650 | 0.8650 | 0.9550 | 0.9250 | 0.8303 | 0.7470 | 0.8615 |
| TKNN-Shap 2023+ | 0.8744 | 0.7724 | 0.5292 | 0.8350 | 0.9300 | 0.9170 | 0.8005 | 0.7190 | 0.8069 |
| KNN-Beta Shap 2022+ | 0.9644 | 0.8468 | 0.6014 | 0.9450 | 0.9000 | 0.9350 | 0.9180 | 0.7319 | 0.7200 | 0.8382 |
| CKNN-Shap (Ours)+ | **0.9814** | **0.8874** | **0.6990** | **0.9700** | **0.9100** | **0.9800** | **0.9450** | **0.8830** | **0.7800** | **0.8929** |
| KNN-Shap 2019a* | 0.9726 | 0.8536 | 0.6482 | 0.9650 | 0.8600 | 0.9450 | 0.9250 | 0.8452 | 0.7620 | 0.8641 |
| KNN-Shap-JW 2023b* | 0.9726 | 0.8536 | 0.6482 | **0.9700** | 0.8600 | 0.9550 | 0.9250 | 0.8463 | 0.7600 | 0.8656 |
| TKNN-Shap 2023* | 0.8378 | 0.7130 | 0.4716 | 0.8400 | 0.8300 | 0.9100 | 0.9180 | 0.7982 | 0.7100 | 0.7810 |
| KNN-Beta Shap 2022* | 0.9622 | 0.8486 | 0.6010 | 0.9550 | 0.9000 | 0.9500 | 0.9130 | 0.7466 | 0.7150 | 0.8435 |
| CKNN-Shap (Ours)* | **0.9814** | **0.8882** | **0.7174** | **0.9700** | **0.9100** | **0.9750** | **0.9450** | **0.8968** | **0.7990** | **0.8981** |

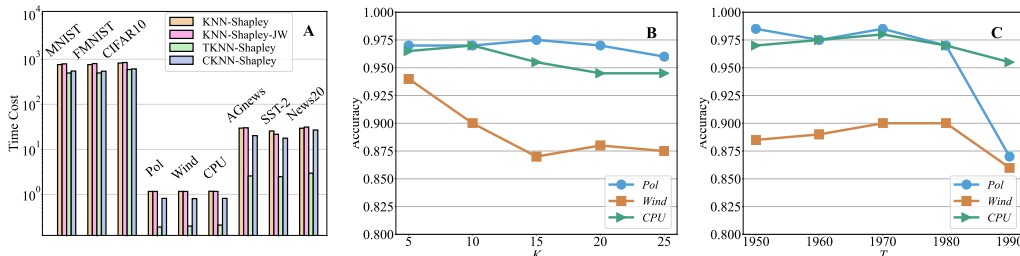

Figure 3: Execution time and parameter analysis. **A** shows the execution time by second in the logarithm of three KNN-Shapley-based data valuation approaches ( KNN-Beta Shapley and KNN-Shapley have identical execution times); **B** and **C** display the classification performance trend of our CKNN-Shapley with different values of $K$ and $T$.

Figure 2. Each dataset is divided into 100 bins, and the goal is to find the intersection of the red and green lines to determine the threshold for distinguishing detrimental samples. The misidentification ratio is then calculated as the proportion of samples in the green region to all detrimental samples. The entire training dataset is segmented into 100 equal-size bins based on the ascending order of data valuation, denoted as $\nu_i$ for the data valuation of the $i$-th bin. Let $p_i$ represent the KNN classifier's performance on the training set without samples in the $i$-th bin, and $p_0$ represent the performance on the entire training set. We define the threshold $t$ for distinguishing detrimental samples from beneficial ones and the misidentification ratio $r$ of detrimental samples as follows:

$$t = \nu_{j^*} \text{ and } r = (j^* - i^*)/j^*, \quad (4)$$

where $j^* = \min_j\{p_j < p_0 \ \& \ p_{j+1} < p_0\}$ and $i^*$ is the index of $i^*$-bin with $\nu_{i^*} = 0$.

**Algorithmic Performance**. We present the algorithmic comparison of different KNN-Shapley-based methods, focusing on value inflation and classification performance. Table 1 displays the thresholds for distinguishing detrimental from beneficial samples and the misidentification ratio of detrimental samples. KNN-Shapley, KNN-Shapley-JW, TKNN-Shapley, and KNN-Beta Shapley exhibit thresholds far from zero, spanning a wide range across different datasets from 0.1632 to 6.4458, from 0.8203 to 4.6543, from 0.7143 to 5.6960, and from 0.0149 to 9.5856, respectively. This variability complicates the interpretation of their data valuation. In contrast, our CKNN-Shapley consistently achieves thresholds close to zero, enhancing the meaningfulness of data valuation. The misidentification ratios of detrimental samples provide additional insights into these thresholds in terms of sample proportions. KNN-Shapley exhibits substantial misidentification ratios, surpassing 50% on 6 out of 9 datasets. KNN-Shapley-JW also surpasses 50% on 3 out of 9 datasets. TKNN-Shapley shows slightly better performance in terms of the threshold but still exceeds 41% inflation on *CPU*. KNN-Beta Shapley, which employs the opposite strategy to our CKNN-Shapley, surpasses 100% performance across all datasets, while CKNN-Shapley with calibrated thresholds, averages only around 10% inflation of detrimental sets and achieves no inflation on *Pol*, *Wind*, and *CPU*.

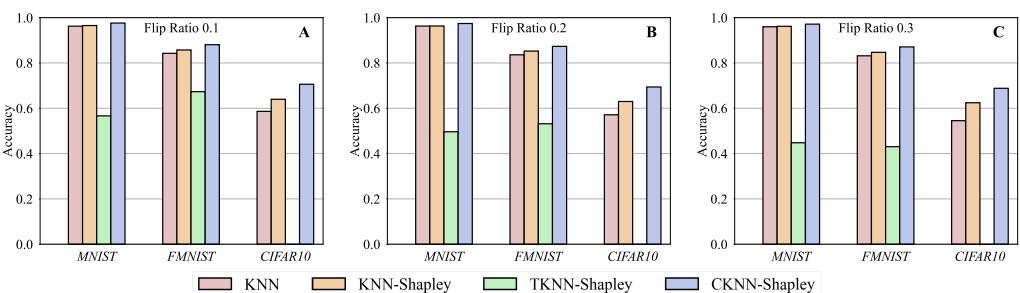

Figure 4: The classification performance of KNN on datasets *MNIST*, *FMNIST*, and *CIFAR10* varies with different training sets and flip ratios. The standard KNN utilizes the full training set, including mislabeled data, whereas KNN-Shapley-based methods start by excluding samples having negative Shapley values from the training set, and then apply the KNN classifier.

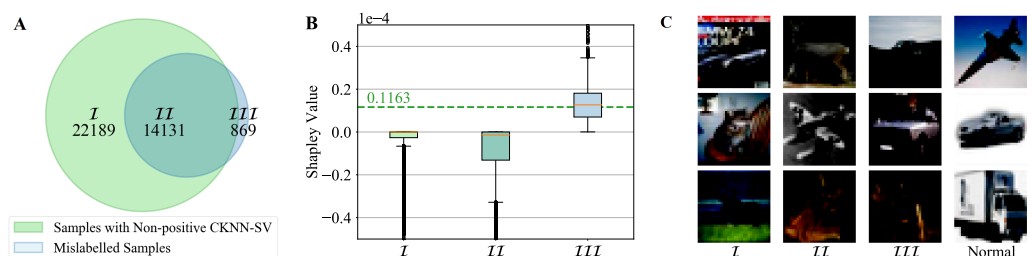

Figure 5: In-depth exploration of CKNN-Shapley on *CIFAR10* with 0.3 flip ratio. **A** depicts the sizes of detrimental and mislabelled samples, where $\mathcal{I}$ denotes the set of samples with non-positive Shapley values but not mislabeled, $\mathcal{II}$ presents the set of joint detrimental and mislabeled samples, and $\mathcal{III}$ is the set of mislabeled samples with positive Shapley values. **B** shows the value distribution of these three sets. **C** displays the visual examples of detrimental or mislabeled samples and normal samples.

In addition to evaluating value inflation, we analyze the impact of inflation on classification performance by removing samples with negative valuation and using weighted KNN in Table 2. KNN-Shapley and KNN-Shapley-JW consistently improve performance on all datasets by removing samples with negative KNN-Shapley values, outperforming vanilla KNN with all training samples. However, TKNN-Shapley, designed for membership protection, does not yield desirable performance. KNN-Beta Shapley does not assign any negative Shapley values across all datasets; thus, its performance is identical to that of vanilla KNN. In contrast, CKNN-Shapley mitigates the negative effects of inflation on detrimental sets, providing additional performance boosts on almost every dataset. Notably, on *CIFAR10* and *SST-2*, CKNN-Shapley exhibits over 7% and 8% performance improvements over KNN-Shapley, addressing the 49% and 56% inflation in misidentification ratios observed in KNN-Shapley on these datasets. In addition to detrimental set inflation, Figure 2 reveals inflation in the beneficial set. We further conduct experiments with a weighted KNN classifier, using weights derived from data valuations, and observe similar phenomena. CKNN-Shapley achieves significant improvements with calibrated data valuations, underscoring the need to address value inflation issues. To address the potential impact of the number of removed samples, we further investigate the setting of removing a fixed amount of samples, as shown in Table 3. Our CKNN-Shapley consistently demonstrates superior performance over various KNN-Shapley-based methods. We provide additional experiments that explore misidentified samples and demonstrate the generalization of our CKNN-Shapley approach to other classifiers in Appendix C.

Figure 3**A** shows the execution time of four methods. TKNN-Shapley has the fastest speed due to its linear time complexity; For KNN-Shalpley, KNN-Shapley-JW, KNN-Beta Shapley and our CKNN-Shapley, all of them have the $\mathcal{O}(N \log N)$; CKNN-Shapley is faster since $T/N$ percentage valuations are directly assigned to zero without the recursive calculation. Figure 3**B** and **C** display the classification performance trend of our CKNN-Shapley with different values of $K$ and $T$, where a small $K$ and large $T$ but not close to $N$ are preferred. A small $K$ excludes the samples from different categories. Besides, a large $T$ enforces the similar to the original one, while maintaining the diversity of selected training subset. We posit that the choice of $T$ is contingent upon the dataset characteristics. Appendix C explores various settings of $T$ and ascertains that $N-2K$ serves as a suitable setting.

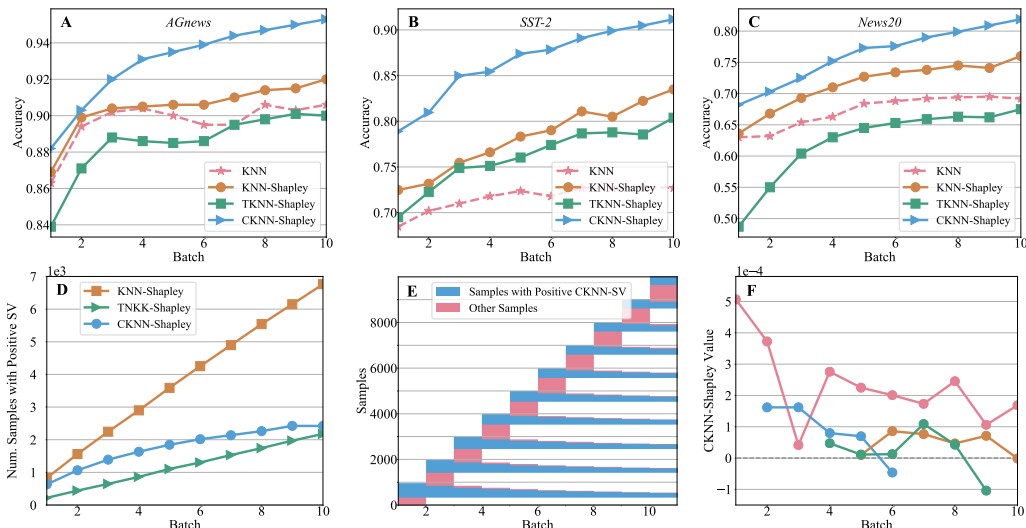

Figure 6: Performance of online learning. **A-C** depict the online learning performance of KNN-Shapley-based data valuation approaches by removing samples with negative Shapley values on the three text datasets across different batches, where the dashed lines present the performance of vanilla KNN without removing any samples. **D** displays the remaining training samples in each batch, combining samples from the last batch and new arrivals in the current batch by removing detrimental samples. **E** provides details of involved training samples for CKNN-Shapley on *News20* across different batches, followed by the Shapley value changes of some representative samples in **F**.

## 5 APPLICATIONS BEYOND CONVENTIONAL CLASSIFICATION

We extend the conventional classification in the previous section by conducting extensive experiments for various practical scenarios, and demonstrate the effectiveness of our proposed CKNN-Shapley in resisting mislabeled data, mitigating distribution shift, and identifying beneficial samples.[3]

**Learning with Mislabeled Data**. We manually simulate mislabeled data by randomly flipping its original label into another category and conduct experiments on *MNIST*, *FMNIST*, and *CIFAR10* with different flip ratios. In Figure 4, we observe the KNN performance with different training sets, where vanilla KNN runs on the complete training set with mislabeled data, and KNN-Shapley-based methods first remove samples with negative Shapley values from the training set before training the KNN classifier. In general, KNN-Shapley and CKNN-Shapley effectively resist the negative impact of mislabeled data, outperforming vanilla KNN. Unfortunately, TKNN-Shapley performs poorly, significantly worse than vanilla KNN, and provides no meaningful results on *CIFAR10*. Due to CKNN-Shapley addressing the value inflation of KNN-Shapley, its performance is further improved and consistently achieves the best results across all three datasets. Notably, CKNN-Shapley maintains similar performance across different flip ratios, demonstrating its resilience to mislabeled data.

Furthermore, we delve into the details of CKNN-Shapley on *CIFAR10* with a 0.3 flip ratio, as shown in Figure 5. In Figure 5**A**, we examine the Shapley values of mislabeled samples, separating the detrimental and mislabeled sets into three categories: $\mathcal{I}$ denotes the set of samples with non-positive Shapley values but not mislabeled, $\mathcal{II}$ represents the set of joint detrimental and mislabeled samples, and $\mathcal{III}$ is the set of mislabeled samples with positive Shapley values. We observe that the majority of mislabeled samples are associated with negative Shapley values, indicating their detrimental nature. Figure 5**B** presents the value distribution of the three sets, revealing a small portion of mislabeled data with very small positive values. From Table 1, we find that the threshold for distinguishing detrimental and beneficial samples on *CIFAR10* is around $0.11 \times e-4$. Several visual examples in Figure 5**C** illustrate the significant distinction between samples $\mathcal{I}, \mathcal{II}$, and $\mathcal{III}$ and normal samples. Samples in $\mathcal{I}, \mathcal{II}$, and $\mathcal{III}$ have a dark background, making them difficult to recognize.

---

[3]We omit KNN-Shapley-JW in this section due to its similar performance with KNN-Shapley. Additionally, we exclude KNN-Beta Shapley since it rarely produces negative valuations.

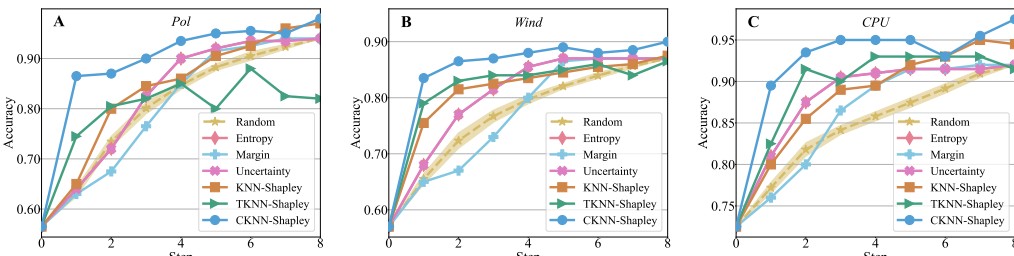

Figure 7: Active learning of different annotation strategies on *Pol*, *Wind*, and *CPU*, where the initial stage has 400 samples and each subsequent step annotates 200 samples into the training set.

**Online Learning**. In addition to static datasets, we further evaluate our approach in the context of online stream data. Specifically, we use three text datasets—*AGnews*, *SST-2*, and *News20*—in this experiment. We segment these datasets into 10 equal-sized batches and gradually feed them into the KNN classifier to simulate stream data. KNN-Shapley-based data valuation approaches are employed to remove detrimental samples with negative Shapley values. Figures 6**A-C** depict the online learning performance of three KNN-Shapley-based data valuation approaches on the three text datasets across different batches. CKNN-Shapley consistently outperforms other methods with large margins, resolving the value inflation issue and showing more effectiveness in identifying detrimental samples in the stream context. Figure 6**D** displays the remaining training samples in each batch, combining samples from the last batch and new arrivals in the current batch by removing detrimental samples. KNN-Shapley involves more training samples compared to others, indicating severe inflation issues. In contrast, CKNN-Shapley removes a significant portion of detrimental samples, enhancing learning performance with fewer training samples. Figure 6**E** provides details of involved training samples for CKNN-Shapley on *News20* across different batches. Additionally, Figure 6**F** shows the Shapley value change for representative samples. The pink line represents a sample included in all batches, while other lines represent samples initially associated with positive Shapley values in the initial batches, which later become negative and are subsequently removed.

**Active Learning**. We further evaluate our approach in the context of active learning. Specifically, to select the unlabeled data for labeling, we use a 2-layer fully connected neural network to fit the relationship between features and Shapley values of labeled training data. Then we use that neural network to predict Shapley values of unlabeled training data and choose the data with the highest predicted Shapley values to label. In this experiment, we choose three chemical datasets —*Pol*, *Wind*, and *CPU*. We segment these datasets into two parts, 20 percent of them are used as labeled data, other 80 percent are used as unlabeled data. The unlabeled data will be taken out 8 times through the prediction of neural network. Figure 7 shows the performance of three KNN-Shapley-based data valuation approaches and three baselines of active learning (i.e., random sampling, entropy sampling (Holub et al., 2008), margin sampling (Balcan et al., 2007), uncertainty sampling (Nguyen et al., 2022)) on three chemical datasets. Random sampling involves the completely random selection of unlabeled samples for labeling; entropy sampling chooses samples with high prediction uncertainty, measured by entropy; margin sampling focuses on the gap between the two highest probabilities in the model's predictions; Uncertainty sampling prioritizes annotation of samples with high nearest neighbor inconsistency. From Figure 7, CKNN-Shapley outperforms other methods and is more effective in identifying useful samples in the unlabeled data, extending the application context of Shapley value and providing a new way to select data for active learning.

## 6 CONCLUSION

In this paper, we revealed the value inflation of KNN-Shapley, which not only misidentifies a large portion of detrimental samples as beneficial, but also overestimates the value for the majority of samples. To address these issues, we proposed Calibrated KNN-Shapley to calibrate zero as the threshold for distinguishing detrimental samples from beneficial ones, via mitigating the negative effects of small training subsets when calculating data valuation. Through extensive experiments, we demonstrated the effectiveness of CKNN-Shapley in alleviating data valuation inflation and detecting detrimental samples. Furthermore, we extended our approach beyond conventional classification settings to the context of learning with mislabeled samples, online learning, and active learning.

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

APPENDIX

## A  ADDITINAL ANALYSIS OF CKNN

Here we provide another understanding of Eq. (3). Take the definition of the KNN utility $U$ in $U(\mathcal{S}) = \frac{1}{K}\sum_{k=1}^{\min\{K,|\mathcal{S}|\}} \mathbb{1}[y_{\alpha_k}=y_v]$, define another utility function $\bar{U}$ by letting $\bar{U}(S) = U(S \cap (D \setminus E))$ where $E = \{z_{\alpha_N}, z_{\alpha_{N-1}}, \cdots, z_{\alpha_{N-T}}\}$. Then, the Shapley value of $\bar{U}$ is exactly Eq. (3).

## B  EXPERIMENTAL SETUP

### B.1  DATASET

Here we use 9 datasets for empirical evaluation. *MNIST*, *FMNIST* (Xiao et al., 2017), and *CIFAR10* are image datasets with 50,000 samples; *Pol*, *Wind*, and *CPU* (Wang et al., 2023) are from the telecommunication, meteorology, and computer hardware domains, respectively, with 2,000 samples each; *AGnews*, *SST-2* (Socher et al., 2013), and *News20* (Lang, 1995) are text datasets with 10,000 samples. For the *CIFAR10* and text datasets, we employ the ResNet50 (He et al., 2016) and Sentence Bert (Reimers and Gurevych, 2019) as embedding for the KNN classifier, respectively. Features/embeddings of these nine datasets are from 14 to 2,048. A comprehensive list of datasets and sources is summarized in Table 4. Following Wang et al. (2023), the validation data size we use is also 10% of the training data size. For *Pol*, *Wind*, and *CPU* datasets, we subsample the dataset to balance positive and negative labels. For the image dataset CIFAR10, we apply a ResNet50 (He et al., 2016) that is pre-trained on the ImageNet dataset as the feature extractor. This feature extractor produces a 1024-dimensional vector for each image. We also employ Sentence BERT (Reimers and Gurevych, 2019) as embedding models to extract features for the text classification dataset *AGNews*, *SST-2*, and *News20*. This feature extractor also produces a 1,024-dimensional vector for each text instance. This method of using a foundation model as a feature extractor can make CKNN-SV well applied to data valuation of deep learning. For other datasets, we do not use any extractor.

Table 4: A summary of datasets used in experiments

| Dataset | # Sample | #Dimension | #Class | Embeddings |
|---------|----------|------------|--------|------------|
| *MNIST* | 50000 | 784 | 10 | None |
| *FMNIST* | 50000 | 784 | 10 | None |
| *CIFAR10* | 50000 | 2048 | 10 | ResNet50 |
| *Pol* | 2000 | 48 | 2 | None |
| *Wind* | 2000 | 14 | 2 | None |
| *CPU* | 2000 | 21 | 2 | None |
| *AGnews* | 10000 | 384 | 4 | Sentence BERT |
| *SST-2* | 10000 | 384 | 2 | Sentence BERT |
| *News20* | 10000 | 384 | 2 | Sentence BERT |

### B.2  BASELINE METHODS

For the baseline methods, we choose the KNN-Shapley (Jia et al., 2019a), KNN-Shapley-JW (Wang and Jia, 2023b), and TKNN-Shapley (Wang et al., 2023) with default setting $K=10$ and $\tau=-0.5$. We also implement Beta Shapley (Kwon and Zou, 2022) in KNN form, called KNN-Beta Shapley with Beta(16,1), which follows the opposite strategy to our CKNN-Shapley, giving significantly higher weight to smaller subsets. Besides, we set $T=N-2K$ in our CKNN-Shapley.

## C  ADDITIONAL EXPERIMENTS

We provide additional experiments of our CKNN-Shapley in terms of exploration of misidentified samples, generalization on other classifiers, and various values of $T$.

## C.1 Correlations between KNN-Shapley and CKNN-Shapley

Table 5 presents the Spearman and Kendall Tau correlation metrics across nine datasets. For datasets such as *MNIST*, *FMNIST*, *CIFAR10*, *AGnews*, *SST-2*, and *News20*, both correlations are very high, indicating nearly identical rankings between KNN-Shapley and CKNN-Shapley. For *Pol*, *Wind*, and *CPU*, while the correlations remain positive, indicating a consistent positive relationship between the rankings of KNN-Shapley and CKNN-Shapley, the correlation strength is comparatively weaker. This difference is further highlighted in Table 3, which shows a clear distinction between CKNN-Shapley and KNN-Shapley when a fixed number of samples are removed based on their rankings.

Table 5: Spearman and Kendall Tau correlation metrics of KNN-Shapley and CKNN-Shapley across different datasets. The Spearman and Kendall Tau metrics measure the strength and direction of the association between the rankings produced by KNN-Shapley and CKNN-Shapley. Positive values indicate a positive correlation, where higher values denote stronger agreement between the rankings of the two methods. Negative values, if present, would indicate an inverse relationship. Larger positive values reflect stronger similarity in the rankings.

| Method\Datasets | MNIST | FMNIST | CIFAR10 | Pol | Wind | CPU | AGnews | SST-2 | News20 |
|---|---|---|---|---|---|---|---|---|---|
| Spearman correlation | 0.9999 | 0.9999 | 0.9999 | 0.6375 | 0.6207 | 0.4955 | 0.9998 | 0.9992 | 0.9999 |
| Kendall Tau correlation | 0.9985 | 0.9988 | 0.9977 | 0.4964 | 0.4825 | 0.3835 | 0.9884 | 0.9791 | 0.9969 |

## C.2 Misidentified Samples

Our CKNN-Shapley method exhibits significantly lower false positives compared to other prevalent methods in Table 6. Since the over-identification of samples as detrimental (FP) is a primary source of value inflation in data valuation, the number of FP samples is much larger than the number of FN samples in all three methods.

Table 6: False positive and false positive of misidentified samples

| Method\Datasets | MNIST | FMNIST | CIFAR10 | Pol | Wind | CPU | AGnews | SST-2 | News20 | Sum. |
|---|---|---|---|---|---|---|---|---|---|---|
| False positive | | | | | | | | | | |
| KNN-Shapley | 47 | 21 | 30 | 68 | 71 | 72 | 52 | 55 | 45 | 409 |
| TKNN-Shapley | 17 | 37 | 0 | 47 | 62 | 64 | 15 | 7 | 14 | 263 |
| CKNN-Shapley | 14 | 9 | 7 | 21 | 28 | 22 | 22 | 8 | 17 | 148 |
| False negative | | | | | | | | | | |
| KNN-Shapley | 0 | 1 | 0 | 1 | 1 | 0 | 2 | 0 | 1 | 6 |
| TKNN-Shapley | 41 | 5 | 55 | 11 | 6 | 8 | 11 | 19 | 18 | 174 |
| CKNN-Shapley | 0 | 5 | 1 | 1 | 2 | 2 | 0 | 2 | 2 | 15 |

## C.3 Generalization

Traditional Shapley values have no constraint on the base classifier, but have to require retraining the base classifier $2^N$ times. Therefore, even for a small dataset, $2^N$ times model training is inflexible for the traditional Shapley values. KNN-Shapley and its variants including our CKNN-Shapley work a series of pragmatic tools computing Shapley values efficiently in a recursive manner with $O(N \log N)$ or $O(N)$ time complexity. For practice, if the base classifier is not KNN, a routine solution employs the KNN classifier as a surrogate.

To test the generalization of our CKNN-Shapley on different models, we conduct extra experiments with non-KNN classifiers, including Multilayer Perceptron (MLP), Logistic Regression (LR), and Support Vector Machine (SVM). Specifically, we train the non-KNN classifiers with the whole training set, apply KNN-Shapley variants to identify detrimental samples, and finally retrain the non-KNN classifiers with the identified detrimental samples removed. Table 7 reports the accuracy on three datasets. We can see that our CKNN-Shapley can effectively boost the performance of non-KNN classifiers in most cases, indicating a good generalization of CKNN-Shapley across different models.

## C.4 Rationality of $T$

We consider two aspects when setting $T$. On one aspect, we believe the value inflation comes from the subset with too small sizes, which leads the sizes of subsets to be close to the whole training

Table 7: Generalizability on other classifier

| Method\Datasets | Pol | Wind | CPU |
|---|---|---|---|
| Vanilla MLP | 0.9909 | 0.8735 | 0.9475 |
| MLP with negative KNN-Shapley value samples removed | 0.9949 | 0.8939 | 0.9455 |
| MLP with negative TKNN-Shapley value samples removed | 0.8515 | 0.8269 | 0.9085 |
| MLP with negative CKNN-Shapley value samples removed | **0.9915** | **0.8979** | **0.9669** |
| Vanilla LR | 0.8700 | 0.8500 | 0.9300 |
| LR with negative KNN-Shapley value samples removed | 0.8800 | 0.8800 | 0.9400 |
| LR with negative TKNN-Shapley value samples removed | 0.8250 | 0.8250 | 0.9200 |
| LR with negative CKNN-Shapley value samples removed | **0.8850** | **0.9350** | **0.9600** |
| Vanilla SVM | **0.9650** | 0.8750 | 0.9400 |
| SVM with negative KNN-Shapley value samples removed | 0.8650 | 0.8850 | 0.9350 |
| SVM with negative TKNN-Shapley value samples removed | 0.8450 | 0.8250 | 0.8900 |
| SVM with negative CKNN-Shapley value samples removed | 0.9600 | **0.8850** | **0.9450** |

set (Large $T$). On another aspect, we expect the subsets to be diverse and the number of subsets to be large enough (Small $T$). By considering both, we give a default setting of $T = N - 2K$, which makes the subset size large enough and consider $2^{N-2K} \times (2^{2K} - 1)$ different subsets.

Moreover, we provide extra experiments of our CKNN-Shalpey with different values of $T$ below. In Table 8, we set $T$ to be $0.95N, 0.75N, 0.5N$ and report the accuracy performance of removing samples with negative values (Larger value means better performance), threshold for distinguishing detrimental from beneficial samples (Closer to zero value means better performance), and misidentification ratio of detrimental samples (Smaller value means better performance). In general, the setting with $N - 2K$ achieves the best average performance on all three metrics. For accuracy, the setting with $N - 2K$ achieves the best performance on all datasets compared with other settings. For other metrics, the setting with $N - 2K$ delivers competitive results. Therefore, we chose $T = N - 2K$ as the default setting.

Table 8: Performance of CKNN-Shapley with different values of $T$

| Method\Datasets | MNIST | FMNIST | CIFAR10 | Pol | Wind | CPU | AGnews | SST-2 | News20 | Avg./Abs. |
|---|---|---|---|---|---|---|---|---|---|---|
| T | 0.9630 | 0.8444 | 0.5956 | 0.9400 | 0.8700 | 0.9200 | 0.9060 | 0.7270 | 0.6920 | 0.8287 |
| Acc after remove samples with negative Shapley values | | | | | | | | | | |
| N-2K | **0.9828** | **0.8884** | **0.7164** | **0.9700** | **0.9000** | **0.9700** | **0.9420** | **0.8980** | **0.7960** | **0.8960** |
| 0.95N | 0.9698 | 0.8444 | 0.6556 | 0.9700 | 0.8850 | 0.9650 | 0.9270 | 0.8612 | 0.7700 | 0.8719 |
| 0.75N | 0.9696 | 0.8444 | 0.6520 | 0.9650 | 0.8600 | 0.9450 | 0.9250 | 0.8463 | 0.7610 | 0.8632 |
| 0.5N | 0.9630 | 0.8444 | 0.6498 | 0.9650 | 0.8650 | 0.9500 | 0.9250 | 0.8474 | 0.7590 | 0.8631 |
| Threshold for distinguishing detrimental from beneficial samples by $\times 1e-4$ | | | | | | | | | | |
| N-2K | **0.0152** | **0.0109** | 0.1163 | -0.3059 | **-0.0245** | **-2.9412** | 1.1955 | **0.1604** | 0.6715 | **0.6046** |
| 0.95N | 0.3888 | 0.2097 | 0.1389 | 0.3889 | 3.6552 | 1.1062 | **1.1398** | 1.7674 | 1.2325 | 1.1141 |
| 0.75N | 0.1970 | 0.2222 | 0.1294 | **-0.1216** | 0.4348 | 3.8879 | 1.7923 | 0.6662 | 1.8334 | 1.0316 |
| 0.5N | 0.2352 | 0.2177 | **0.0718** | 0.4385 | 4.3249 | 2.4663 | 1.7139 | 0.8520 | 1.7043 | 1.3361 |
| Misidentification ratio of detrimental samples | | | | | | | | | | |
| N-2K | 0.2000 | **0.1686** | **0.0908** | **0.0000** | **0.0000** | **0.0000** | **0.1538** | **0.1143** | **0.1821** | **0.1011** |
| 0.95N | **0.1250** | 0.3333 | 0.3736 | 0.1250 | 0.0833 | 0.2727 | 0.5172 | 0.2895 | 0.4737 | 0.2811 |
| 0.75N | 0.6325 | 0.4566 | 0.3600 | 0.0000 | 0.1000 | 0.4615 | 0.5493 | 0.2500 | 0.5800 | 0.3767 |
| 0.5N | 0.7107 | 0.4385 | 0.3404 | 0.1111 | 0.1429 | 0.3333 | 0.6739 | 0.2432 | 0.5680 | 0.3985 |

# D LIMITATIONS AND BROADER IMPACT

Our research focuses on overcoming challenges associated with value inflation in the application of KNN-Shapley for data valuation influence estimation. By introducing a method that enables accurate assessment of training samples' impact on model performance, we provide a tool for practitioners to determine the positive or negative effects of these samples. Through comprehensive testing across various problem scenarios, our Shapley value removal strategy has been proven superior to existing methods, enhancing model efficiency by eliminating harmful data points. Consequently, our contributions have the potential to drive substantial societal benefits, particularly as the use of more complex and expansive neural networks, like Large Language Models, becomes more prevalent. For the limitations, we do not provide a theoretical analysis on the selection of $T$, where we posit that the choice of $T$ is contingent upon the dataset characteristic. To tackle this limitation, we recommend $T = N - 2K$ as the default setting for its satisfactory performance on the datasets in our experiments.

# E    CODE AND REPRODUCIBILITY

We provide our code, instructions, and implementation in an open-source repository: https://anonymous.4open.science/r/Inflation_KNN-SV-D9C1.

All experiments were conducted on a workstation with an AMD Ryzen Threadripper PRO 5965WX CPU and x86_64 architecture with 128 GB memory, using NVIDIA GeForce RTX 4090 GPUs with 24GB VRAM running CUDA version 12.3 and driver version 545.23.08.

