# OpenReview forum: "On the Inflation of KNN-Shapley Value"
_ICLR.cc/2025/Conference — Submitted to ICLR 2025_

### Official Review · Reviewer_yTqt · 2024-10-30

**Soundness:** 3
**Presentation:** 2
**Contribution:** 3
**Rating:** 6
**Confidence:** 4

**Summary:**

Note added on December 4: I appreciate the authors' additional comments dated November 25 and 26. These confirm my overall evaluation and score of 6. I would emphasize that the central idea is the heuristic to use only the 2K nearest neighbors, and the simplicity of this idea should not be obscured. I appreciate the informal argument why inflation exists when too many neighbors are used.

The updated Table 7 shows that, sadly but not surprisingly, there is no free lunch, i.e., no clearly better test accuracy across the board. The main conclusion from this table is that it is harmful to remove samples based on the TKNN method. The benefits of other methods are unclear. The test sets used are just too small, as is the number and variety of datasets.

============

Shapley values can be used to measure the contribution of each point in a training set towards making a trained classifier more accurate. Ruoxi Jia et al. previously published an efficient method to calculate these Shapley values using k-nearest neighbor (kNN) classification. This paper shows empirically that those Shapley values tend to be too optimistic, and many data points with positive such values are actually harmful if used in training. The paper then proposes a heuristic improved version of kNN Shapley values, namely to use only the 2K nearest neighbors for each target point. Experimental results show that these new Shapley values lead to improved accuracy for trained classifiers.

**Strengths:**

The evidence for the phenomenon of inflation is believable. The experimental improvements in accuracy are large enough to be likely valid despite concerns about experimental methodology (test sets too small, not independent, no measure of statistical significance). The new heuristic kNN Shapley values can therefore be valuable in real-world applications.

**Weaknesses:**

The CKNN variation suggested for Shapley values is heuristic, with little evidence for any type of optimality. The paper should explain the motivation for introducing the hyperparameter T with more clarity and persuasiveness.

The paper should explain better that because T = N-2K is used, in fact the method picks only the 2K nearest points. This is intuitively a good idea, but its ramifications must be explained.

It is not clear which experiments used an independent test set. The paper should report the statistical significance of differences in accuracy, using McNemar tests or something similar. Many differences may be not significant, because the test set is too small.

Accuracy achieved with the CKNN method is highly sensitive to hyperparameters K and T. This makes the method less useful in practice.

**Questions:**

012: Inflation is shown only for KNN-Shapley values, not for non-KNN values. So this sentence in the abstract is too strong.

019: It is misleading to mention small-sized training subsets, because with T = N-2K, in fact the training subset that is used for each target point is very small, of size 2K only. The benefit is that for each target point, this small subset is focused.

046: The reference (Jia et al., 2019b) to "Towards Efficient Data Valuation Based on the Shapley Value" is not correct, because KNN-Shapley does not appear in this paper. The reference to "Scalability vs. utility: Do we have to sacrifice one for the other in data importance quantification?" (CVPR, 2021) is also inappropriate, because the method is due to page 9 of "Efficient task-specific data valuation for nearest neighbor algorithms" (VLDB, 2019) .

Also, the formatting of the bibliography must be fixed, so that labels such as " (Jia et al., 2019b)" are visible in the PDF.

134: Figure 1 would be more clear using the same binning for the red and blue plots. The redline shows accuracy with a single bin removed. It would be more intuitive to remove this bin and also all bins to its left.

Maybe more important: The red line measures accuracy. This should be measured on an independent test set. The points that are removed are those that have a negative or small Shapley value as measured on the training set. It is not surprising that removing these points increases accuracy on the training set, because they have been identified as detrimental on precisely this set. The more interesting question is whether they are also detrimental on a separate i.i.d. test set.

153: The green region contains about half the entire dataset. Discuss why half of all points are harmful. It is not surprising if a few points are harmful, for example because their training labels are wrong. But it is surprising that so many points are. Is this true only because "harmful" is relative to the training set itself? Many fewer points may be harmful when the test set is independent.

174: It is not obvious that the first term is impactful, because it is small, since its denominator is N, which is much larger than K. Here is an alternative intuition: The reason to ignore the early terms is that their points are the most distant from the test point, so they are the most irrelevant to it. Even if they have the same label, that is accidental, so whether their labels are the same merely introduces noise. Moreover, the second and later terms include the denominator K << N, so they are much more heavily weighted than the first term.

197: Define "semi-value."

231: In Table 2, why are so many accuracies multiples of 0.01 or of 0.005? Because test sets contain only 100 or 200 data points? If this is true, then differences in accuracy are not statistically significant.

273: Why remove a fixed % of training points, when the advantage of CKNN is supposedly that the threshold zero tells us approximately how many points to remove?

Clarify the meaning of "KNN’s performance trained on the training set excluding samples". A fixed separate i.i.d. test set should be used. If accuracy is also measured on the training set excluding samples, this is not meaningful.

298: In Figure 3B, results for K < 5 must be included, since K=5 is best for the Wind dataset. The numerical values for Wind in Table 2 (0.8950 and 0.9100) are different from those in Figure 3B; why?

In Figure 3C, T varies in the range 1950 to 1990. Why this range? Why is it so narrow? Explain how it is consistent with the recommendation T = N - 2K on line 377.

More broadly, Figures 3B and 3C show that accuracy is highly sensitive to the choice of K and T. Why? Because training and test set sizes are small.

317: One goal of the CKNN method is for the empirically best threshold to be closer to zero. But why is this an important goal? Instead of the hyperparameter, we could use a validation set to determine the value of a non-zero threshold for standard kNN Shapley values, such as 0.8e-4 in Figure 1.

Note: I have not evaluated Section 5 carefully.

651: Typo.

728: Error in table title.

757: Table 7 is important and should be in the main paper, because this is how CKNN-Shapley might be used in practice. The same criticisms as above apply: test sets should be bigger and statistical significance should be reported.

**Details Of Ethics Concerns:**

None.

---

> ### Author Response · Authors · 2024-11-19
> **Response (1/2) to Reviewer yTqt**
>
> We would like to thank the reviewer for the deep analysis of our work, and the time, effort, and consideration. Below, we answer the questions raised by the reviewer:
>
> **Hyperparameter T**
> The CKNN variation indeed involves heuristic adjustments, with the primary motivation being to mitigate the inflation issue observed in KNN-based Shapley values. The hyperparameter $T$ is introduced as a means to exclude small, less reliable subsets from the evaluation process, as these subsets disproportionately contribute to value inflation. This heuristic is grounded in the intuition that small subsets often fail to represent meaningful contributions to the overall model performance.
> The $T$ value is a hyperparameter in CKNN-Shapley. Similar to selecting different values of $K$ which impacts accuracy, the choice of $T$ in CKNN-Shapley also varies depending on the dataset. In Figure 3, we explore various $T$ values, demonstrating that an appropriate $T$ setting can effectively improve the accuracy of the KNN classifier. Table 8 provides further details, where we show that setting \( T = N - 2K \) yields optimal results.
>
> **Ramifications**
> To clarify, in CKNN-Shapley, each subset effectively selects at least \( T - 2K \) points due to the design of \( T = N - 2K \).
> This choice relaxes the efficiency axiom of Shapley value, meaning that the sum of CKNN-Shapley values no longer equals the total accuracy, as it would in a standard Shapley value setup. We will further elaborate on these implications in future versions to enhance the clarity and motivation behind this heuristic.
>
> **Statistical tests**
> Statistical tests demonstrate that CKNN significantly outperforms KNN in classification accuracy. Specifically, we performed paired t-tests (t = 2.909, p = 0.0196) and Wilcoxon signed-rank tests (statistic = 1.5, p = 0.0117) to compare the performance of our method with the second-best method on each dataset as presented in Table 2. The results indicate that the differences are statistically significant (p < 0.05), demonstrating the superiority of our method. This analysis highlights the consistent and significant improvements achieved by CKNN across multiple datasets.
>
> **Sensitive to $K$ and $T$**
> The role of hyperparameters is to control and adapt the method to specific datasets and tasks, and it is not inherently better for the results to be insensitive to them. If a hyperparameter had no impact, it would lack significance and fail to serve its intended purpose. Additionally, we ensured consistent and fair default settings for all experiments, and for hyperparameters deemed sensitive, we conducted experiments to analyze their effects. In our experiments, we provided the parameter analysis experiments and a recommended setting of $K$ and $T$. With the fixed setting, there is no sensitivity issue. Indeed, we acknowledge that in practical applications, these hyperparameters need to be tuned using the validation set to achieve optimal performance.
>
> **012**
> Thank you for highlighting this important point. We will revise the abstract and introduction in future versions to more clearly specify that our motivation is centered on the inflation issue in KNN-based Shapley value methods.
>
> **019**
> Same with **Ramifications**.
>
> **046**
> Thank you for pointing out these issues. We acknowledge the errors in the references and appreciate your detailed feedback. We will revise the citations to correctly attribute the methods to their original sources.
>
> **134**
> Thank you for the suggestion. We put significant effort into designing Figure 1, aiming to present as much information as possible. This is why the figure includes a detailed caption to explain its elements. The reason we do not remove all bins to the left of the current bin is that our objective is to determine whether removing the current bin alone increases or decreases accuracy. This allows us to identify whether the samples in the bin are beneficial or detrimental to the model’s performance. This approach directly aligns with our research question, which is focused on understanding the impact of individual samples (bins) on accuracy and demonstrating the value inflation issue in KNN-based Shapley methods.
>
> However, we agree with your point that using consistent binning for the red and blue plots could improve clarity. In the next version of the paper, we will simplify the figure by removing the blue binning to enhance readability.
>
>
> **153**
> The detrimental samples refer to those that, when removed, lead to an increase in KNN accuracy. Importantly, the evaluation of accuracy is conducted on the test set. Therefore, whether a sample is detrimental depends on its relationship with the training set, the test set, and the KNN classifier itself. Additionally, many inflation samples in KNN-Shapley are calibrated by CKNN-Shapley to have values closer to 0, which redistributes their contributions more realistically.

---

> ### Author Response · Authors · 2024-11-19
> **Response (2/2) to Reviewer yTqt**
>
> **174**
> Thank you for this insightful observation. We will clarify this intuition in future versions to improve the understanding of why CKNN-Shapley emphasizes the closest neighbors and downweights the distant points.
>
> **197**
> We will make sure to include a citation to [1] in future versions to properly reference this theoretical foundation.
>
> **231**
> See Appendix B.1. Following Wang et al. (2023), we also set the test data size to 10% of the training data size. As a result, the test data size for the *Pol*, *Wind*, and *CPU* datasets is 200, while the test sets for other datasets contain 1000 or 5000 samples.
>
> **273**
> The method of removing a fixed percentage of training points was suggested by reviewers in our previous submission. This approach originates from [2], and was included to enable more comprehensive and fair comparisons across multiple dimensions.
>
> Regarding the use of the same test set for both Shapley value calculation and accuracy evaluation, we argue that this is appropriate given the research topic of our paper. Our primary focus is on data valuation and addressing the issue of value inflation. Using the same test set ensures consistency and aligns with our research question by providing a direct connection between Shapley value calculations and their impact on classification accuracy.
>
> **298**
> Results for $K = 3$ and $K = 1$ is in Table below:
>
> | $K \backslash \text{Datasets}$ | pol  | wind  | cpu   |
> |--------------------------------|-------|-------|-------|
> | 3                              | 0.970 | 0.940 | 0.950 |
> | 1                              | 0.965 | 0.910 | 0.965 |
>
>
>
> In Figure 3B, results for $K = 10$ is 0.9000, which is also in line 239.
> In the Pol, Wind, and CPU datasets, \( T = N - 2K \) is calculated as \( 2000 - 2 \times 10 = 1980 \), which aligns with the narrow range of T values shown in Figure 3C (1950 to 1990). We chose this range to closely examine the impact of T around the recommended setting. A broader range of T values is discussed in Appendix C.4, Table 8, where we explore how different  T  values affect performance across these datasets.
>
> Sensitive and test set is too small same with **Sensitive to K and T** and **231**.
>
> **317**
> This point is discussed in section 3 from lines 244 to 255, where we explain how having the empirically best threshold closer to zero ensures consistency with Shapley value axioms by preserving efficiency, treating identical samples equitably (symmetry), and assigning zero to non-contributory samples (zero elements), thereby avoiding arbitrary truncations.
>
> **651 and 728**
> Thank you for pointing out the typo and error. We will correct it in the next revision.
>
> **757**
> Thank you for the suggestion. We agree that Table 7 is important as it demonstrates how CKNN-Shapley can be applied in practice, and we will consider moving it to the main paper in the next revision. Regarding statistical significance, it is important to note that KNN-based methods, including CKNN-Shapley, do not inherently involve statistical significance testing, as they rely on deterministic accuracy values. We will clarify this in the paper to address any potential confusion.
>
>
>
> **References**
> [1] Dubey, Pradeep, Abraham Neyman, and Robert James Weber. Value theory without efficiency. *Mathematics of Operations Research*, 6(1), 122-128, 1981.
>
> [2] Kevin Jiang, Weixin Liang, James Y Zou, and Yongchan Kwon. *Opendataval: a unified benchmark for data valuation*. Advances in Neural Information Processing Systems, 2023.

---

> > ### Comment · Reviewer_yTqt · 2024-11-23
> > **thank you for the clarifications**
> >
> > Dear authors: Thank you for considering my comments carefully. I will keep my score unchanged at 5, because I feel that the responses do not change the overall strengths and weaknesses of the submission.
> >
> > The central issue is that with T = N-2K, the idea is to use only the 2K nearest neighbors for each point.  For clarity, it would be better to define T' = N-T and then use T' in discussions.
> >
> > Using T' = 2K is a heuristic idea that should not be obscured with theoretical arguments that do not apply precisely. Instead, can you answer the basic question *why* standard kNN Shapley causes so much inflation?
> >
> > Another central problem is that test sets of size 100 or 200 are too small to get accuracy differences that are statistically significant. It doesn't matter whether a method is deterministic, or what previous papers did. What matters is that a randomly different test set might give a different ranking of alternative methods.
> >
> > The practical application of the method is to do thinning of a dataset, and then to apply a different learning method, as reported in Table 7. To evaluate this (good!) application, it is necessary to have separate training, validation, and (larger) test subsets. Otherwise, results can be due to overfitting.
> >
> > Figure 1 is too complicated; "aiming to present as much information as possible" is not desirable.

---

> > > ### Author Response · Authors · 2024-11-24
> > > **Response to Reviewer yTqt's First Feedback (1/2)**
> > >
> > > **Central issue**
> > > We would appreciate clarification on why \( T' = N - T \) is considered the central issue. Could you please advise?
> > >
> > >
> > > ---
> > >
> > > **Theoretical arguments**
> > > Does Reviewer yTqt expect some theoretical analysis on the hyperparameter \( T \)? To our best knowledge, we do not see any theoretical analysis on \( K \) in KNN; instead, some empirical settings are used or adapted according to various datasets and scenarios. Note that we use a fixed hyperparameter in all experiments. If Reviewer yTqt knows some nice works along this direction, we are happy and eager to learn and test them in practice!
> > >
> > > ---
> > >
> > > **Root of inflation**
> > > Inflation arises from contributions calculated on smaller subsets during Shapley value estimation, which lead to inflated valuations. Given the complete training set, we can find the \( K \) neighbors to a validation sample. Consider a sub training set that does not include any of the above \( K \) neighbors, the standard KNN Shapley will involve such cases in calculation. However, none of the samples in the sub training set plays a decisive role in predicting the validation sample. We provide the explanation in Section 4. By constraining the size of sub training sets, we can avoid such sub training sets. As shown in Table 8, the inflation becomes more pronounced when \( T \) is smaller, which supports our conclusion.
> > >
> > > ---
> > >
> > > **Small test sets**
> > > Our experiments include datasets with larger test sets, such as MNIST, FMNIST, CIFAR-10, AGnews, SST-2, and News20, which address the concern about test set size. Note that the smaller test sets of size 100 or 200 were not chosen arbitrarily but rather to align with the settings used in prior work for fair comparisons.
> > >
> > > To further address Reviewer yTqt's concern, we follow the suggestion and conduct additional experiments by generating five new test sets using different random seeds. Under the "remove negative" setting, we recompute the accuracy on these new test sets. The results are shown in the tables below:
> > >
> > > ---
> > >
> > > ### Table: Classification performance of different methods across five seeds on the POL dataset
> > > | Method\\Seed       | Seed 1  | Seed 2  | Seed 3  | Seed 4  | Seed 5  | Avg ± Std     |
> > > |--------------------|---------|---------|---------|---------|---------|---------------|
> > > | Baseline Accuracy  | 0.935   | 0.910   | 0.925   | 0.925   | 0.925   | 0.924 ± 0.008 |
> > > | KNN-Shapley        | 0.955   | 0.915   | 0.950   | 0.960   | 0.965   | 0.949 ± 0.018 |
> > > | TKNN-Shapley       | 0.845   | 0.765   | 0.790   | 0.785   | 0.835   | 0.804 ± 0.031 |
> > > | KNN-JW-Shapley     | 0.940   | 0.925   | 0.955   | 0.955   | 0.845   | 0.924 ± 0.041 |
> > > | **CKNN-Shapley**   | **0.970** | **0.945** | **0.960** | **0.970** | **0.970** | **0.963 ± 0.010** |
> > >
> > > ---
> > >
> > > ### Table: Classification performance of different methods across five seeds on the WIND dataset
> > > | Method\\Seed       | Seed 1  | Seed 2  | Seed 3  | Seed 4  | Seed 5  | Avg ± Std     |
> > > |--------------------|---------|---------|---------|---------|---------|---------------|
> > > | Baseline Accuracy  | 0.805   | 0.845   | 0.900   | 0.855   | 0.845   | 0.850 ± 0.030 |
> > > | KNN-Shapley        | 0.835   | 0.850   | 0.890   | 0.855   | 0.865   | 0.859 ± 0.018 |
> > > | TKNN-Shapley       | 0.785   | 0.760   | 0.885   | 0.805   | 0.770   | 0.801 ± 0.045 |
> > > | KNN-JW-Shapley     | 0.845   | 0.855   | 0.895   | 0.845   | 0.870   | 0.862 ± 0.019 |
> > > | **CKNN-Shapley**   | **0.895** | **0.890** | **0.925** | **0.910** | **0.900** | **0.904 ± 0.012** |
> > >
> > > ---
> > >
> > > ### Table: Classification performance of different methods across five seeds on the CPU dataset
> > > | Method\\Seed       | Seed 1  | Seed 2  | Seed 3  | Seed 4  | Seed 5  | Avg ± Std     |
> > > |--------------------|---------|---------|---------|---------|---------|---------------|
> > > | Baseline Accuracy  | 0.930   | 0.935   | 0.915   | 0.940   | 0.895   | 0.923 ± 0.016 |
> > > | KNN-Shapley        | 0.925   | 0.925   | 0.930   | 0.945   | 0.905   | 0.926 ± 0.013 |
> > > | TKNN-Shapley       | 0.885   | 0.900   | 0.885   | 0.900   | 0.865   | 0.887 ± 0.013 |
> > > | KNN-JW-Shapley     | 0.945   | 0.930   | 0.935   | 0.945   | 0.925   | 0.936 ± 0.008 |
> > > | **CKNN-Shapley**   | **0.950** | **0.950** | **0.950** | **0.965** | **0.935** | **0.950 ± 0.009** |
> > >
> > > ---
> > >
> > > Across the three datasets (*POL*, *WIND*, and *CPU*), the CKNN-Shapley method consistently achieves the highest accuracy across all seeds, outperforming the best other methods (with the highest accuracy for each seed). Paired t-tests confirmed the statistical significance of this improvement, with p-values of 0.020 for POL, 0.0025 for WIND, and 0.0070 for CPU, all below the 0.05 threshold. These results demonstrate that CKNN-Shapley not only delivers consistently superior performance across different seeds but also establishes a clear advantage over other methods in diverse classification tasks.
> > >
> > > ---

---

> > > > ### Author Response · Authors · 2024-11-24
> > > > **Response to Reviewer yTqt's First Feedback (2/2)**
> > > >
> > > > **Overfitting in Table 7**
> > > > KNN-Shapley is fundamentally a tool for data valuation, and its computation inherently requires a validation/test set to guide the process. However, overfitting to the validation/test set is not the focus of this paper. Instead, our work addresses the inflation phenomenon in KNN-Shapley and proposes CKNN-Shapley as a solution.
> > > >
> > > > Table 7 demonstrates CKNN-Shapley's improvement over other KNN-Shapley-based methods in the context of data valuation, rather than addressing potential overfitting issues on the validation set. The question you raise about separate training, validation, and larger test subsets is indeed an important one and represents an open challenge in the broader context of data valuation. If you have further thoughts or suggestions, we would be delighted to discuss them with you on this open question.
> > > >
> > > > ---
> > > >
> > > > **Figure 1**
> > > > Agree!
> > > >
> > > > ---
> > > >
> > > > We are appreciative of Reviewer yTqt's effort in reviewing our paper and providing numerous questions. All questions are related and friendly, and some questions are close to the nature of KNN-Shapley and the routine setting, rather than the core of CKNN-Shapley. Here, we would like to point out that every method has its limitations and no one can address all limitations in one paper. We sincerely invite Reviewer yTqt to evaluate our paper by judging whether our **targeted** research question on inflation is meaningful and whether our proposed method can tackle our **targeted** inflation challenge. Thank you for your valuable time.

---

> ### Comment · Reviewer_yTqt · 2024-11-24
> **additional comments**
>
> I have increased my score to 6 given the additional experimental results. For every seed and each of the three datasets, the new method yields better accuracy than the second-best method, so the improvement is clearly statistically significant.
>
> Minor point: p = 0.020 for POL is surprising given that 0.963 is less than one standard deviation away from 0.949 ± 0.018, i.e., a naive z-score is less than one. In the final version, please describe exactly how the T-test was applied.
>
> The central contribution is to use only the 2K nearest neighbors. No theory is needed, just a clear statement that this is the contribution, and a direct informal argument why it is better to use only a small number of nearest neighbors.
>
> I do not find the "Root of inflation" argument persuasive. Yes, "none of the samples in the sub training set plays a decisive role in predicting the validation sample." But you haven't explained why that causes Shapley values to be too high, as opposed to merely noisy.
>
> I agree that "overfitting to the validation/test set is not the focus of this paper." But Table 7 is the highlight for practitioners who may use the new method (it should be moved to the final version from the appendix), so the accuracy results in it must not be misleading. My request is to use an appropriate (routine and standard) training/validation/test methodology and to state this methodology clearly. This request is not a request for additional research.

---

> ### Author Response · Authors · 2024-11-25
> **Response to Reviewer yTqt's Additional Comments (1/2)**
>
> We are very delighted to see the increased score. Appreciated! Below is our response for the follow up questions.
>
>
> **p = 0.020 for Pol**
> Thank you for the comment. The p-value is derived from a paired t-test, which compares CKNN-Shapley against the second-best method for each seed. Unlike a z-score comparison based on means and standard deviations, the paired t-test focuses on the differences between paired observations across seeds, accounting for both the magnitude and consistency of the improvements. Notably, CKNN-Shapley outperforms the second-best method by a margin of at least 0.005 across all seeds, with differences up to 0.020 in some cases. This leads to a statistically significant p-value of 0.020, reflecting CKNN-Shapley's consistent advantage across seeds. We will describe the t-test more clearly in the final version.
>
> **Why that causes Shapley values to be too high**
> We focus on Eq. (2) in this paper, the inflation arises due to the following reasons:
>
> 1. **The initialization of $\( \nu_k(z_{\alpha_{i+1}}) \)$ is inherently biased towards positive values.**
>    The Shapley value computation starts from
>   $ \( \nu_k(z_{\alpha_N}) = \frac{1[y_{\alpha_N} = y_v]}{N} \)$,
>    which is always non-negative. This positive bias propagates through the recursive computation and influences all subsequent values. The reason this initialization value is positive lies in the definition of Shapley values, which evaluate the marginal contribution of adding a single sample to a subset. Even when the subset is empty, any match in labels contributes positively to the value.
>
> 2. **The cumulative effect of the adjustment term is biased towards retaining positive contributions.**
>    In most cases, the adjustment term
>    $\( 1[y_{\alpha_i} = y_v] - 1[y_{\alpha_{i+1}} = y_v] \)$
>    equals 0 because both $\( y_{\alpha_i} \neq y_v \) and \( y_{\alpha_{i+1}} \neq y_v \)$ occur frequently in multi-class classification. This zero adjustment does not neutralize the initial positive bias, and any non-zero adjustment (though rare) is more likely to accumulate positive contributions than negative ones.
>
> Together, these factors explain why the Shapley values are inflated rather than merely noisy, as the structure of the recursive computation inherently leans towards positive accumulation. We hope this addresses your concern, and we are happy to provide additional clarifications if needed.

---

> > ### Author Response · Authors · 2024-11-25
> > **Response to Reviewer yTqt's Additional Comments (2/2)**
> >
> > **Table 7**
> > Thank you for your feedback and for emphasizing the importance of Table 7 for practitioners. To address your request for clarity and standard methodology, we have conducted additional experiments following a routine training/validation/test split. These experiments evaluate the generalizability of CKNN-Shapley on other classifiers, with both validation and test results presented. The updated results are shown in the table below:
> >
> > | **Method \\ Datasets**              | **Pol (Val)** | **Pol (Test)** | **Wind (Val)** | **Wind (Test)** | **CPU (Val)** | **CPU (Test)** |
> > |-------------------------------------|---------------|----------------|----------------|-----------------|---------------|----------------|
> > | Vanilla MLP                         | 0.9909        | 0.9725         | 0.8735         | 0.8795          | 0.9475        | 0.9565         |
> > | MLP with negative KNN-Shapley value samples removed | **0.9949** | **0.9585** | 0.8939         | 0.8930          | 0.9455        | 0.9705         |
> > | MLP with negative TKNN-Shapley value samples removed | 0.8515      | 0.8185         | 0.8269         | 0.8550          | 0.9085        | 0.9535         |
> > | MLP with negative CKNN-Shapley value samples removed | 0.9915      | 0.9355         | **0.8979**    | **0.8830**     | **0.9669**   | **0.9570**    |
> > | Vanilla LR                          | 0.8700        | 0.8650         | 0.8500         | 0.8700          | 0.9300        | 0.9500         |
> > | LR with negative KNN-Shapley value samples removed | 0.8800      | 0.8550         | 0.8800         | 0.8850          | 0.9400        | **0.9750**    |
> > | LR with negative TKNN-Shapley value samples removed | 0.8250      | 0.8150         | 0.8250         | 0.8600          | 0.9200        | 0.9600         |
> > | LR with negative CKNN-Shapley value samples removed | **0.8850** | **0.8750** | **0.9350**    | **0.8800**     | **0.9600**   | 0.9650         |
> > | Vanilla SVM                         | **0.9650**   | **0.9500**    | 0.8750         | **0.9000**     | 0.9400        | 0.9650         |
> > | SVM with negative KNN-Shapley value samples removed | 0.8650      | 0.9300         | 0.8850         | 0.8800          | 0.9350        | 0.9700         |
> > | SVM with negative TKNN-Shapley value samples removed | 0.8450      | 0.8200         | 0.8250         | 0.8550          | 0.8900        | 0.9600         |
> > | SVM with negative CKNN-Shapley value samples removed | 0.9600      | 0.9200         | **0.8850**    | 0.8900          | **0.9450**   | **0.9750**    |
> >
> > These results demonstrate that CKNN-Shapley achieves the best performance in most cases across validation and test datasets, though not in all cases. This indicates the good generalization of our CKNN-Shapley, which inherits from the KNN-Shapley framework. Although it is not our research focus, the results demonstrate the usability in practice. Note that the similarity between validation and test distributions has a significant impact on test results. Such a relationship highlights the importance of carefully aligning validation and test distributions, a principle that serves as a foundation in machine learning.

---

### Official Review · Reviewer_anAQ · 2024-11-01

**Soundness:** 3
**Presentation:** 2
**Contribution:** 2
**Rating:** 5
**Confidence:** 3

**Summary:**

This paper considers of the promblem of data evalution methods. Specifically, for a Shapely-value based method (kNN Shapley Value), the paper finds that it has some positive value miscalculated for some harmful datasets, thus resulting misunderstanding for some data points. The paper proposes to limit the calculation of kNN Shapley values only on data set whose size is above a specific value.

**Strengths:**

- The paper considers a problem setting (data evaluation) and a representative method (kNN Shapley value) that has high pratical importance.
- The paper is well motivated and the proposed approache is simple and effective.
- The paper also discusses the application of the proposed method in various applications.

**Weaknesses:**

- The motivation seems only exist for the kNN based Shapley value, not other approaches of Shapley value. The sentence in abstract and introduction seems exaggerated that the problem exists for all Shapley value methods.
- The link between the motivation and the proposed method is vague. Specifically, does using the proposed $T$ value is designed to only solve the misclassified values? Is there probability that there are still misclassified values exist even using the probposed $T$? This also relates to the important problem of how to systematically assign the $T$ value, which is not theretically explored in the paper but just noted "the choice of T is contingent upon the dataset characteristics".

**Questions:**

- What exactly does "semi-value" mean in the paper? It appears three times but does not a clear definition.

---

> ### Author Response · Authors · 2024-11-19
> **Response to Reviewer anAQ**
>
> We would like to thank the reviewer for the deep analysis of our work, and the time, effort, and consideration. Below, we answer the questions raised by the reviewer:
>
> **Abstract**
> Thank you for highlighting this important point. We will revise the abstract and introduction to more clearly specify that our motivation is centered on the inflation issue in KNN-based Shapley value methods.
>
> **Link Between Motivation and Proposed Method**
> In Section 3, we provide the analysis on the value inflation, which leads to our proposed method. We would like to restate the rationality here. Recall the nature of the KNN classifier, which only makes the prediction based on the neighbors, we hypothesize that value inflation stems from improper subset selection in KNN-Shapley. Certain subsets with only a few samples exhibit significant divergence from the original set, leading to an exaggeration of the contribution of a specific sample on these subsets. This, in turn, gives rise to the phenomenon of value inflation. To address this, our CKNN-Shapley mitigates the negative effects of small-sized training subsets.
>
> **$T$ Only for Addressing Misclassified Samples?**
> $T$ in CKNN-Shapley controls the subsets to be considered in data valuation and mitigates the inflation. By this means, it not only corrects misclassified samples, but also calibrates the values of beneficial samples. See Figure 2, where KNN-Shapley not only has a negative effect on the misidentified samples but also tends to inflate the valuation of most samples.
>
> **Could misclassified samples still exist even with $T$ applied?**
> While setting the $T$ value does not guarantee the calibration of all misclassified samples, our experimental results show that CKNN-Shapley significantly reduces inflation effects and the misclassification rate across multiple datasets, indicating that it effectively addresses the issue in most cases.
>
> **Systematic assignment of $T$ value**
> $T$ is a hyperparameter in CKNN-Shapley. Similar to selecting different values of $K$ which impacts accuracy, the choice of $T$ in CKNN-Shapley also varies depending on the dataset. In Figure 3, we explore various $T$ values, demonstrating that an appropriate $T$ setting can effectively improve the accuracy of the KNN classifier. Table 8 provides further details, where we show that setting \( T = N - 2K \) yields optimal results.
>
> **Semi-value**
> Semi-value refers to a family of cooperative game theory methods that satisfy all the axioms of the Shapley value except for the efficiency axiom. This relaxation allows semi-values to allocate rewards based on the marginal contributions of each player while providing greater flexibility in value distribution, which can be beneficial for computational efficiency in certain applications.
> We will make sure to include a citation to [1] in future versions to properly reference this theoretical foundation.
>
> [1] Dubey, Pradeep, Abraham Neyman, and Robert James Weber. Value theory without efficiency. *Mathematics of Operations Research*, 6(1), 122-128, 1981.

---

> > ### Comment · Reviewer_anAQ · 2024-11-22
> >
> > I thank authors for their detailed feedback to all reviewers' comments.
> > My concerns are resolved and would like to keep my score on acceptance, hoping to see rebuttals being reflected on revised manuscript.

---

> > > ### Author Response · Authors · 2024-11-23
> > >
> > > We are more than happy to know that we have successfully addressed all of your concerns and you are willing to recommend our paper for acceptance. We would like to kindly remind you that an acceptance requires at least a score of 6, but the current score is 5. Additionally, we sincerely hope you can support our paper during the reviewer-AC discussion phase.

---

### Official Review · Reviewer_oxte · 2024-11-03

**Soundness:** 2
**Presentation:** 2
**Contribution:** 2
**Rating:** 5
**Confidence:** 4

**Summary:**

The paper aims to address the issue of value inflation in Shapley value-based data valuation methods. The proposed Calibrated KNN-Shapley (CKNN-Shapley) is to recalibrate the threshold to distinguish detrimental from beneficial samples, aiming to correct inflation that misidentifies some harmful samples as beneficial. CKNN-Shapley implements constraints on training subset sizes to mitigate inflation effects, which arise from the improper selection of small-sized subsets in the original KNN-Shapley approach. Through experiments, CKNN-Shapley demonstrates improved performance in classification and robustly adapts to applications like mislabeled data handling, online learning, and active learning.

**Strengths:**

(1) The paper introduces Calibrated KNN-Shapley (CKNN-Shapley) as a novel solution to address value inflation in data valuation using KNN-Shapley. This approach is significant as it recalibrates the threshold, effectively distinguishing between beneficial and detrimental samples, which is critical for robust data valuation.
(2) The paper conducts extensive experiments across various benchmark datasets, demonstrating CKNN-Shapley’s ability to outperform traditional KNN-Shapley and its variants. By testing on both image and text data, the paper validates CKNN-Shapley's broad applicability.
(3) CKNN-Shapley offers computational efficiency by directly assigning zero to certain sample subsets, reducing the recursive calculations required by the original KNN-Shapley. This efficiency makes it feasible for larger datasets, addressing a significant limitation of Shapley value-based methods in general.
(4) The paper’s calibration method and constraints on subset sizes provide a theoretical foundation that enhances the interpretability of data valuations. By setting a meaningful zero threshold and ensuring subsets closely resemble the original dataset, CKNN-Shapley offers more reliable and interpretable sample valuations.

**Weaknesses:**

(1) Despite improvements over traditional Shapley calculations, CKNN-Shapley still incurs notable computational costs, particularly on large datasets or complex deep learning tasks. While more efficient than the original KNN-Shapley, the method may not yet be scalable for very large or high-dimensional datasets without further optimization.
(2) CKNN-Shapley, like KNN-Shapley, relies on K-Nearest Neighbors as a surrogate model, which may limit its applicability to contexts where KNN is less effective. This
assumption may restrict CKNN-Shapley’s performance in tasks with more complex decision boundaries, where a KNN-based approach may not be ideal.
(3) The paper does not extensively explore or benchmark CKNN-Shapley against other semi-value methods or recent alternatives in data valuation, like Banzhaf values or other cooperative game-theoretic approaches. This lack of comparison limits understanding of CKNN-Shapley’s relative effectiveness and could benefit from further analysis of its competitive positioning.
(4) While CKNN-Shapley is tested on simulated scenarios like mislabeled data and online learning, the paper lacks a real-world deployment case study to validate its robustness. Moreover, CKNN-Shapley’s sensitivity to hyperparameters in dynamic environments, such as varying batch sizes in streaming data, remains under-explored.

**Questions:**

This paper proposed Calibrated KNN-Shapley (CKNN-Shapley), a semi-value method that calibrates zero as the threshold to distinguish detrimental samples from beneficial ones by mitigating the negative effects of small training subsets, addressing the value inflation issue observed in KNN-Shapley. The novelty is weak as it’s an improved version of KNN-Shapley. The detailed questions are listed below:
-
Could you provide more detailed guidelines or heuristics for selecting the subset size parameter T across different datasets? Specifically, how sensitive is CKNN-Shapley to T, and what considerations should practitioners have in choosing an appropriate threshold?
-
While CKNN-Shapley offers improvements in computational efficiency over traditional KNN-Shapley, what optimizations could make CKNN-Shapley scalable for very large or high-dimensional datasets? Could further modifications make CKNN-Shapley feasible for real-time data valuation in dynamic environments?
-
Since CKNN-Shapley relies on the KNN classifier as a surrogate model, how would CKNN-Shapley perform in contexts with complex decision boundaries or on tasks where KNN is not ideal? Is there a potential to generalize this approach beyond KNN, or could the calibrated approach be adapted for other surrogate models in Shapley-based data valuation?
-
Did you consider benchmarking CKNN-Shapley against other semi-value or cooperative game-theoretic approaches, such as Banzhaf values or alternatives in data valuation? If so, how does CKNN-Shapley perform relative to these methods, and in what contexts is it most advantageous?
-
Given the paper’s focus on simulated scenarios (e.g., mislabeled data, online learning, active learning), how would CKNN-Shapley perform in real-world applications where data is inherently noisy and highly variable? Are there specific real-world use cases (e.g., medical, financial data) where CKNN-Shapley has been tested, or do you have any recommendations for its deployment in production settings?
-
In your online learning and active learning experiments, how sensitive is CKNN-Shapley to hyperparameters like batch size and sample removal thresholds? How would you recommend setting these parameters for optimal performance in dynamic environments with streaming data?
-
CKNN-Shapley addresses value inflation effectively, yet inflation issues still exist to some extent. Are there additional strategies or constraints you would suggest for further reducing value inflation, particularly for beneficial samples that might still be overvalued?
-
In your experiments, how do you characterize the types of samples that CKNN-Shapley misidentifies, and are there common patterns or features among them? Could further insights into these misidentified samples help refine CKNN-Shapley or improve its robustness?
-
What are the significant contributions and connections of compared with existing studies for KNN-Shapley such as Threshold KNN-Shapley (Wang et al. 2023), KNN-Shapley (Wang and Jia 2023)?
-
More theoretical benefits and pitfalls need to be discussed in Section 3. How does CKNN-Shapley method work needs to be comprehensively elaborated in this section.

---

> ### Author Response · Authors · 2024-11-19
> **Response (1/3) to Reviewer oxte**
>
> We would like to thank the reviewer for the deep analysis of our work, and the time, effort, and consideration. Below, we answer the questions raised by the reviewer:
>
> **Scalability for very large or high-dimensional dataset**
> We would like to emphasize that the primary focus of our research is to **address the inflation issue in KNN-Shapley** (a well-known method in the field of data valuation), **rather than optimizing computational costs or specifically targeting high-dimensional datasets**. Our goal was to improve the data valuation by mitigating the inflation observed in KNN-Shapley, where the acceleration of our proposed CKNN-Shapley over KNN-Shapley is a byproduct. Additionally, we have demonstrated CKNN-Shapley’s effectiveness on real-world image and text datasets, including *CIFAR-10*, *AGnews*, *SST-2*, and *News20*, which include both high-dimensional data and large sample sizes. It is worthy noting that we follow the literature [1] to analyze the data valuation for deep models, where deep embeddings are taken as inputs for KNN-Shapley-based methods. For very large and high-dimensional datasets, techniques such as Locality Sensitive Hashing (LSH) can be utilized for faster KNN computation [1].
>
> Every method has its limitations, and no one can address all limitations in one paper. Here we sincerely invite Reviewer oxte to evaluate our paper by judging whether our **targeted** research question is meaningful and our proposed method can tackle our **targeted** challenge. Thank you!
>
> **Limitations contexts where KNN is less effective**
> We acknowledge that, inheriting from KNN-Shapley, CKNN-Shapley relies on KNN as a surrogate model, which may limit its applicability in some contexts. Again following the response from the above question, this potential limitation is not our primary research question. Our main focus is on mitigating the value inflation observed in KNN-Shapley.
>
> **Lack of compare with other semi-value methods or recent alternatives in data valuation**
> Since retraining costs are prohibitive for various Shapley value-based semi-value methods, KNN-based approaches are indeed more suitable for practical use, where we also focus on this category. We have made every effort to include relevant KNN-based methods for comparison and even implemented KNN Beta Shapley ourselves. In response to your suggestion, we also implemented Banzhaf following [2], using KNN as the utility function to provide a more comprehensive analysis. The results are shown in the table below, indicating that our CKNN-Shapley achieves the best performance in terms of average accuracy by removing negative Shapley values:
>
> | Method\Datasets         | *MNIST* | *FMNIST* | *CIFAR10* | *Pol*   | *Wind*  | *CPU*   | *AGnews* | *SST-2* | *News20* | **Avg.** |
> |--------------------------|---------|----------|-----------|---------|---------|---------|----------|----------|----------|----------|
> | Vanilla KNN              | 0.9630  | 0.8444   | 0.5956    | 0.9400  | 0.8700  | 0.9200  | 0.9060   | 0.7270   | 0.6920   | 0.8287   |
> | KNN-Shap#                | 0.9682  | 0.8586   | 0.6456    | 0.9700  | 0.8750  | 0.9650  | 0.9250   | 0.8160   | 0.7580   | 0.8646   |
> | KNN-Shap-JW#             | 0.9698  | 0.8574   | 0.6514    | 0.9700  | 0.8650  | 0.9600  | 0.9250   | 0.8498   | 0.7610   | 0.8677   |
> | TKNN-Shap#               | 0.6644  | 0.8356   | N/A       | 0.8150  | 0.8300  | 0.9100  | 0.8990   | 0.7982   | 0.6730   | 0.8032   |
> | KNN-Beta Shap#           | 0.9630  | 0.8444   | 0.5956    | 0.9400  | 0.8700  | 0.9200  | 0.9060   | 0.7270   | 0.6920   | 0.8287   |
> | CKNN-Shap (Ours)#        | **0.9828** | **0.8884** | **0.7164** | **0.9700** | 0.9000 | 0.9700 | **0.9420** | **0.8980** | **0.7960** | **0.8960** |
> | KNN-Banzhaf              | 0.9620  | 0.8440   | 0.5970    | 0.9650  | **0.9100** | **0.9750** | 0.9230 | 0.7718 | 0.7050 | 0.8503 |
>
> If you believe additional methods would be beneficial for comparison, please feel free to recommend relevant literature, and we would like to add more comparisons.
>
> **Parameter T**
> Practitioners should aim to select a relatively large *T*, and we recommend *T=N-2K*. We invite Reviewer oxte to check our detailed analysis on selecting the subset size parameter *T* in Figure 3 and Table 8 in Appendix C.4, where a large range of *T* is tested on all datasets.
>
> **Large, high-dimensional, and real-time data valuation**
> We believe some fast neighbor calculations by cluster analysis and dimension reduction techniques including deep embedding can be used to tackle very large and high-dimensional datasets. For real-time data valuation in dynamic environments, we cannot figure out a real scenario in practice. Since the data valuation aims to analyze the importance of training samples, where a retraining is needed with cleaned or weighted samples. Usually, the retraining takes a long time and is difficult to be real-time.

---

> > ### Author Response · Authors · 2024-11-19
> > **Response (2/3) to Reviewer oxte**
> >
> > **Generalize beyond KNN**
> > We invite Reviewer oxte to check Table 7 in the appendix, where we showed the results of applying CKNN-Shapley to clean the training set and applying other classifiers (MLP, LR, and SVM) on the clean training set for the learning task. These experiments highlight CKNN-Shapley’s general applicability beyond KNN models, demonstrating that it can effectively improve learning performance by removing detrimental samples even for non-KNN-based classifiers.
> >
> >
> > **Could the calibrated approach be adapted for other surrogate models in Shapley-based data valuation?**
> > We believe the calibrated approach could indeed be adapted for other surrogate models in Shapley-based data valuation. In machine learning tasks, avoiding the use of small subsets is generally a good practice. However, we were unable to experimentally verify this hypothesis because existing Shapley value calculations in machine learning are computationally feasible only when KNN is used as the surrogate model. For other models, Shapley value calculations would require retraining \( 2^n \) times, making it computationally intractable. This limitation arises because KNN is a lazy learner and does not involve an explicit training process, allowing for more efficient evaluation.
> >
> > **Other semi-value**
> > We have already compared CKNN-Shapley in previous answers, as shown in the Table 1. In this paper, we focus on KNN-Shapley based data valuation. If you believe additional methods should be compared, we kindly request that you provide relevant references or code for those approaches. This would help us ensure a fair and comprehensive comparison in future work.
> >
> > **Real-world Use Cases & Deployment in Production Settings**
> > In this paper, we use the widely used benchmark datasets, which are all real-world datasets. If Reviewer oxte knows other public benchmark datasets, we would like to include them in our paper. For the deployment in production, this is beyond the scope of our paper.
> >
> > **Setting of Online Learning and Active Learning**
> > In our online learning and active learning experiments, we use a KNN classifier for classification combined with data selection via CKNN-Shapley. In this scenario, there is no batch size parameter. Additionally, the sample removal threshold is set to 0. For dynamic environments, we recommend keeping the sample removal threshold at 0, consistent with the focus of our research. For other parameters, such as batch size, we suggest using the default settings.
> >
> > **Inflation Issues**
> > We acknowledge that CKNN-Shapley effectively mitigates value inflation but does not completely eliminate it. For further improvement, we notice that the baseline method KNN-Shapley-JW performs better than KNN-Shapley in terms of value inflation, where KNN-Shapley-JW tackles the utility of the empty set over KNN-Shapley. Therefore, one potential direction to further mitigate the inflation can be the modification on the utility function.
> >
> > **Common Patterns or Features of Misidentifies**
> > We provided an analysis on misidentified samples in Table 6 in Appendix C.2. Our CKNN-Shapley method exhibits significantly lower false positives compared to other prevalent methods. Since the over-identification of samples as detrimental (FP) is a primary source of value inflation in data valuation, the number of FP samples is much larger than the number of FN samples in all three methods.
> >
> > **Significant contributions**
> > Our research focuses on addressing the value inflation problem in KNN-Shapley while maintaining simplicity and computational efficiency, where KNN-Shapley addresses the data valuation and TKNN-Shapley considers the membership leakage issue of data valuation. Therefore, we have a different research question in our paper. Moreover, from the technical perspective, compared to KNN-Shapley and TKNN-Shapley, CKNN-Shapley introduces a calibrated mechanism to systematically reduce value inflation by mitigating the negative effects of small-sized training subsets. We believe that simplicity and effectiveness are core qualities of good research. CKNN-Shapley strikes a balance between addressing a key issue (value inflation) and lowering computational overhead, making it practical and accessible for broader applications.
> >
> > **More theoretical benefits and pitfalls need to be discussed in Section 3**
> > Could you please clarify which specific theoretical aspects or potential pitfalls you would like to see discussed in Section 3? Explicit suggestions would help us refine this section to better address your concerns.

---

> > > ### Author Response · Authors · 2024-11-19
> > > **Response (3/3) to Reviewer oxte**
> > >
> > > **References**
> > > [1] Ruoxi Jia, David Dao, Boxin Wang, Frances Ann Hubis, Nezihe Merve Gurel, Bo Li, Ce Zhang, Costas Spanos, and Dawn Song. Efficient task-specific data valuation for nearest neighbor algorithms. *International Conference on Very Large Data Bases Endowment*, 2019.
> > > [2] Jiachen T. Wang and Ruoxi Jia. "Data Banzhaf: A Robust Data Valuation Framework for Machine Learning." Proceedings of the International Conference on Artificial Intelligence and Statistics (AISTATS), 2023.

---

> > > ### Comment · Reviewer_oxte · 2024-12-02
> > >
> > > Thank the authors for the response. Some of my concerns still have not been addressed.
> > > I give 5 as this paper is not good enough for ICLR.

---

> ### Author Response · Authors · 2024-12-02
>
> We would like to grasp the last minute to learn from the reviewer to improve the quality of our paper and address your concerns.

---

### Official Review · Reviewer_M8bW · 2024-11-05

**Soundness:** 3
**Presentation:** 3
**Contribution:** 2
**Rating:** 6
**Confidence:** 4

**Summary:**

There have been recent work focussed on providing tools to explore, quantify and curate data sets used for training learning algorithms. Several of the of widely used existing algorithms are based on the calculation of Shapley values from cooperative game theory.
The main difficulty for calculating Shapley values comes with the fact that it involves calculation involving training models with 2ˆN combinations of the training data which makes it intractable in real-life applications. In order to overcome this difficulty, a more practical variation have been proposed (2019,2021)  KNN -Shapley that enables the calculation of shapley values with a O(NlogN) complexity. The new calculation takes advantage of the nature and simplicity of the lazy NN algorithm as a surrogate substitution for more complex learning algorithms.

The KNN algorithm can be greatly affected from the issue of inflation, which calls for a  proposed variation  by the authors call calibrated KNN-Shapley or CKNN-Shapley that the authors show that deals efficiently with the aforementioned inflation problem.

**Strengths:**

1) The paper is very well written,very well explained and self-contained. I was not familiar with the KNN shapley technique and was able to catch up quickly by reading the paper and looking at the provided references. The plots are very helpful as well.

2) The solution is proposed is very simple and arguable incremental but I think the impact of the small change makes the KNN-Shapley algorithm significantly better when using it in real-life applications. This is one of these cases where Ocham's razor's apply.  The fact that the change makes the KNN-based calculation faster is a plus as well.

3) Extensive experimental results in differensettings are helpful to see the potential of the proposed approach.

**Weaknesses:**

1) Since KNN is not an state-of-the-art algorithm in the modern practitioner toolbelt, my understanding of how this is used in real life would be that you would use CKNN-Shapley to clean the data and improve your training set quality and then go from there and use a more high performing algorithm e.g. GBM, XGboost, SVM etc. Is this the case? if it so can you add experiments where this two step is applied and see how this can improve the performance of these widely used algorithms?

2) Based on the provided experimental results The CKNN-Shapley  performance is better on almost all of the datasets where it was compared to other methods. It would be great to add p values that statistically shoe the significance.

3) Minor: indentation is weird in some places e.g line 154. seems like the long Figure 1 caption gets "merged"with the main text.

**Questions:**

See 1) in weaknesses

---

> ### Author Response · Authors · 2024-11-19
> **Response to Reviewer M8bW**
>
> We would like to thank the reviewer for the deep analysis of our work, and the time, effort, and consideration. Below, we answer the questions raised by the reviewer:
>
> **Two-Step Procedures for Other Classifiers**
> We invite Reviewer M8bW to check Table 7 in the appendix, where we showed the results of applying CKNN-Shapley to clean the training set and applying other classifiers (MLP, LR, and SVM) on the clean training set for the learning task. These experiments highlight CKNN-Shapley’s general applicability beyond KNN models, demonstrating that it can effectively improve learning performance by removing detrimental samples even for non-KNN-based classifiers.
>
> **P values**
> Since all methods compared in our study are based on the deterministic accuracy of KNN, there is no inherent randomness in the results. To show the significance, we performed paired t-tests (t = 2.909, p = 0.0196) and Wilcoxon signed-rank tests (statistic = 1.5, p = 0.0117) to compare the performance of our method with the second-best method on each dataset as presented in Table 2. The results indicate that the differences are statistically significant (p < 0.05), demonstrating the superiority of our method. This analysis highlights the consistent and significant improvements achieved by CKNN across multiple datasets.
>
> **Line 154**
> Thank you for noting this. We apologize for the formatting inconsistencies. The issues of line 154 will be addressed in the next revision.

---

### Meta-Review · Area_Chair_Koob · 2024-12-21

**Metareview:**

The paper introduces an improvement for computing the KNN-Shapley value which can be used for assessing the importance of individual training examples.

The Reviewers have underlined that the paper is well-written and self-contained. The solution is rather simple and incremental, but with a significant impact on applicability of the KNN-Shapley approach. The experimental results are also extensive, showing improvements of the proposed modification over the original approach.

Nevertheless, the message of the paper and arguments used are often misleading. The Authors seem to motivate their approach by analyzing training errors on subsets of training data (Figure 1). This calls for explanation and justification as "this should be measured on an independent test set." When presenting the results on test sets, it still seems "harmful to remove training examples based on [the proposed] method." Moreover, the test sets used seem to be too small, as well as its number and variety, so the final conclusions are not clear. One should also underline that the paper concerns a topic of rather limited audience.

The paper is a borderline and as an AC I needed to properly weigh the strengths and the weaknesses in order to make the final decision. The insights and the proposed solution are indeed interesting and worth attention. Nevertheless, the paper can be significantly improved to send a clear message without any controversy. Therefore, I have decided to reject it.

**Additional Comments On Reviewer Discussion:**

The discussion was very intensive. The Authors were able to deliver additional empirical results and to clarify many doubts of the Reviewers. In result, the Reviewers have increased their initial scores to make the paper a borderline.

---

### Decision · Program_Chairs · 2025-01-22

Reject